# A climate index for the Newfoundland and Labrador shelf

Frédéric Cyr[1,2] and Peter S. Galbraith[3]

[1]Northwest Atlantic Fisheries Centre, Fisheries and Oceans Canada, St. John's, NL, Canada
[2]Memorial University of Newfoundland, St. John's, NL, Canada
[3]Maurice Lamontagne Institute, Fisheries and Oceans Canada, Mont-Joli, QC, Canada
**Correspondence:** Frédéric Cyr (Frederic.Cyr@dfo-mpo.gc.ca)

**Abstract.** This study presents in detail a new climate index for the Newfoundland and Labrador (NL) shelf. The NL climate index (NLCI) aims to describe the environmental conditions on the NL shelf and in the Northwest Atlantic as a whole. It consists of the average of 10 normalized anomalies, or subindices, derived annually: winter North Atlantic Oscillation, air temperature, sea ice season severity, iceberg count, seasonal sea surface temperature, vertically-averaged temperature and salinity at the Atlantic Zone Monitoring Program (AZMP) Station 27, summer cold intermediate layer (CIL) core temperature at AZMP Station 27, summer CIL area on 3 AZMP hydrographic sections and bottom temperature on the NL shelf. This index runs from 1951 to 2020 and will be updated annually. It provides continuity in the production of advice for fisheries management and ecosystem status on the NL shelf, for which a similar but recently abandoned index was used. The new climate index and its subindices are available at https://doi.org/10.20383/101.0301 (Cyr and Galbraith, 2020).

## 1 Introduction

Climate indices are often regarded as simple ways to relate the mean environmental conditions to the state of an ecosystem. As changes in the ocean climate affect species distribution and the resilience of the World's ecosystems (Pörtner et al., 2019), these climate indicators are currently in high demand. Useful and accessible climate indices are also keystone pieces of information for fisheries management, especially now that many governments and inter-governmental agencies show an increasing desire to implement an ecosystem approach to their fisheries management (e.g., for the North Atlantic: Levin et al., 2008; Pepin et al., 2019; Koen-Alonso et al., 2019; Muffley et al., 2021).

In eastern Canada, climate indices are often used by Fisheries and Oceans Canada (DFO) in ecosystem studies and/or to inform fisheries scientists and managers as part of various regional assessments of marine resources. At DFO, these indices and related advice are often provided by scientists affiliated to the Atlantic Zone Monitoring Program (AZMP, Therriault et al., 1998). Created in 1998 following a general consensus that *"changes in climate cannot be ignored as an explanation for fluctuations in marine resources"* (from the Introduction of the founding document), the AZMP had a mandate to develop and implement an efficient and effective environmental monitoring program for the northwest Atlantic zone, and then to maintain it through the years. The creation of the AZMP came as a direct consequence of the collapse of the Atlantic Zone groundfish fisheries in the early 1990's, a period characterized by anomalously cold and potentially unfavorable environmental conditions

for the stock, although the latter was highly debated (Lear and Parsons, 1993; de Young and Rose, 1993; Hutchings and Myers, 1994; Mullowney et al., 2019).

Since 2000, annual reports produced by AZMP scientists on the oceanographic and meteorological conditions of the Canadian Atlantic Zone have been published through the Canadian Science Advisory Secretariat (CSAS). The idea of combining several normalized anomalies into a single index representative of the climate of a large area was first develop by Petrie et al.
(2006a) for the meteorological, sea ice and sea surface temperature (SST) parameters in eastern Canada, and by Petrie et al. (2006b) for the physical oceanography of the Scotian Shelf and the Gulf of Maine. These *composite indices* were derived by summing 23 and 18 individual normalized anomalies, respectively (see Table 1 for a review of CSAS publications using such indices). While the physical oceanographic index from the Scotian Shelf and the Gulf of Maine is still produced annually (see Hebert et al., 2020, for latest update), the eastern Canada meteorological, sea ice and SST index was last produced by Petrie
et al. (2009a). Starting with the annual report for 2009 conditions, the meteorological, sea ice and physical oceanographic conditions were integrated into a single report (Hebert et al., 2011, see also Table 1) and the eastern Canada atmospheric and ocean index was abandoned. Note that for the Newfoundland and Labrador (NL) shelf, sea ice and atmospheric conditions were reported together with the physical oceanographic conditions starting with the annual report of Colbourne et al. (2006), although no composite climate index was presented (see Table 1).

Following the approach of Petrie et al. (2006a, b), a *composite climate index* was developed for the NL shelf by Colbourne et al. (2008). This composite climate index was derived by summing 26 individual normalized anomalies (e.g., winter North Atlantic Oscillation, air temperature at several sites, surface and bottom temperature at several sites, average temperature, salinity and area of water colder than 0°C along oceanographic transects, etc.; see Table 1). These individual components did not, however, begin with the same year. The number of components of this composite climate index was increased to 28
in 2015 (Colbourne et al., 2015). Although never explicitly named this way, this composite climate index (both the 26 and the 28 components definitions) rapidly gained popularity among biologists, ecologist and fisheries scientists in NL under the appellation *Composite Environmental Index*, or CEI, the term used hereafter for commodity. The CEI was used, for example, to assess the prospect of several commercial marine resources (Koen-Alonso et al., 2010; DFO, 2014), marine mammals stock fluctuations (DFO, 2020b; Hammill et al., 2020; Stenson et al., 2020) and to assess the ecosystem productivity as a whole
(NAFO, 2017, 2018). The CEI was last produced by Cyr et al. (2019).

A new climate index for the NL shelf is introduced here to ensure the continuity of the studies mentioned above, and in a hope to feed new similar initiatives. This index was first introduced by Cyr et al. (2020), but it is formalized, detailed and made openly accessible for the first time here. At the moment, this index runs from 1951 to 2020, but will be updated annually (see data availability). The main differences with previous versions is the reduced number of components (10) and the fact that
the index is now calculated as the average of the different components rather than their sum. Details are provided in the next section.

## 2 NL climate index components

The NL climate index (NLCI) aims to represent the large-scale climate conditions and state of the physical environment on the NL shelf and the Northwest Atlantic in general (see Figure 1 for map). The index is a composite of 10 components (or
subindices) estimated annually by DFO (e.g. Cyr et al., 2020). These subindices are presented in the following sub-sections.

The NLCI and most subindices start in 1951. Prior to 1951, there are too few observations for many of subindices to achieve a level of accuracy and precision that would provide high confidence in their estimates. Except for the North Atlantic Oscillation (NAO), these sub-indices are presented in terms of normalized anomalies ($\tilde{X}$):

$$\tilde{X} = \frac{X - \overline{X}_{\text{clim}}}{sd(X)_{\text{clim}}}, \tag{1}$$

where $X$ represents any annual time series, and $\overline{X}_{\text{clim}}$ and $sd(X)_{\text{clim}}$ are the average and standard deviation of $X$ over the climatological period. Following recommendations form World Meteorological Organization standards, the climatological period used is 1991-2020 (World Meteorological Organization, 2017). Because this period was updated at the end of 2020, a complete series of relevant figures for this study are also provided in the Appendix referenced to the previous climatological period of 1981-2010.

### 2.1 North Atlantic Oscillation

The North Atlantic Oscillation (NAO) refers to the anomaly in the sea-level pressure (SLP) difference between the sub-tropical high (near the Azores) and the subpolar low (near Iceland). While several operational definitions of the NAO exist, the monthly NAO based on empirical orthogonal functions (EOF) from the National Center for Environmental Information of the National Oceanic and Atmospheric Administration is used here. This definition better represents the larger-scale influence of the SLP
patterns above the Northwest Atlantic, and tends to be less noisy than the station-based definition. This time series is available online (https://www.ncdc.noaa.gov/teleconnections/nao/). Because the monthly NAO index is normalized over the 1950-2000 period, this subindex is not re-normalized here.

The winter NAO, defined as the average of monthly values from December to March, is considered here (Figure 2). A positive phase of the NAO index is usually associated with an intensification of the Icelandic Low and the Azores High SLP.
Except for some years for which the SLP patterns are spatially shifted (e.g., 1999, 2000 and 2018), positive winter NAO years usually favor strong Northwesterly winds, cold air and sea surface temperatures, and heavy ice conditions in the Northwest Atlantic (Colbourne et al., 1994; Drinkwater, 1996; Petrie et al., 2007a). A predominance of strongly positive winter NAO phase has persisted since 2012, including the record high of +1.61 in 2015 (the record low of -1.47 was in 2010). This recent positive phase of the NAO also corresponded to an intensification of the convection in the Labrador Sea (Yashayaev and Loder,
2017).

## 2.2 Air Temperature

The air temperature subindex consists of the average of the annual normalized anomalies at five coastal communities around the Northwest Atlantic: Nuuk (Greenland), Iqaluit (Baffin Island), Cartwright (Labrador) and Bonavista and St. John's in Newfoundland (see Figure 1). These sites were chosen because they are representative of both local (e.g., sea ice formation and SST) and remote (e.g., icebergs originating from Greenland and freshwater fluxes from the Canadian Arctic) effects on the NL shelf. While the data for Nuuk are obtained from the Danish Meteorological Institute (Vinther et al., 2006), the air temperature data from the Canadian sites are from the second generation of Adjusted and Homogenized Canadian Climate Data (AHCCD) that accounts for shifts in station location and changes in observation methods (Vincent et al., 2012). When necessary, they are updated using Environment Canada's National Climate Data and Information Archive.

Annual normalized anomalies since 1951 for these five sites are presented in Figure 3 under the form of a stacked bar plot. Despite the fact that these sites are spread around the North Atlantic, they often exhibit consistency in the sign of their anomalies, especially in the periods of very cold or warm years. While 1972 appears as the coldest year of this time series, the period of the early 1990's exhibits a sustained period of cold air temperature, with 1991, 1992 and 1993 being respectively the sixth, second and third coldest years of the entire time series. This cold period was followed by the predominance of warmer-than-normal air temperatures at all sites from the mid-1990s to about 2013, with 2010 being the warmest year on record by a large extent. Except for 2015, which was the coldest year since 1993, and 2018 (slightly cold), recent years were close to normal.

## 2.3 Sea Ice

Sea ice season duration and maximum cover area are estimated from ice cover products obtained from the Canadian Ice Service (CIS). The methodology is described in Galbraith et al. (2020) and briefly summarized here. The source CIS products consist of weekly Geographic Information System (GIS) charts covering the East coast for the period 1969-2020 and Hudson Bay for the period 1980-2020. The Hudson Bay charts include coverage of the Northern Labrador Shelf and the East coast charts cover the Southern Labrador Shelf as well as the Newfoundland Shelf (Figure 1). For each of the three regions, the seasonal maximum area of sea ice was determined as well as the ice season duration (Figure 4). The former accounts for partial coverage (as opposed to sea ice extent) and the latter is obtained from a spatial average of the number of weeks with sea ice at every pixel, with zeros counted for areas where no ice was present but the 30 year climatology shows some. The normalized anomalies of these 6 time series (duration and area over three regions) are averaged into a single index presented in a stacked bar plot (Figure 4, bottom panel). Each stacked color in this plot represents its respective contribution to the average. The numerical values of this sea ice subindex are reported in a color-coded scorecard at the bottom of this panel. Negative anomalies, indicative of warmer conditions, have been colored red, and positive anomalies blue. This time series corresponds to the sea ice contribution to the NL climate index. The periods of maximum sea ice cover and season duration are found in the early 1970's, mid-1980's and early 1990's. Since the early 1990's the severity of the sea ice season has gradually decreased,

reaching the lowest values in 2011 and 2010. With the exception of a rebound to near-normal values in 2014-2016, sea ice conditions have been weak over recent years.

## 2.4 Iceberg count

The number of icebergs drifting south of 48°N in the Northwest Atlantic (see dashed cyan box in Figure 1) has been monitored by the International Ice Patrol (IIP) of the US Coast Guard since 1900 (International Ice Patrol, 2020). These icebergs mostly originate from western Greenland (Marson et al., 2018) and may carry with them important amount of freshwater (Martin and Adcroft, 2010). The entire time series is presented in Figure 5. The 121-year average annual number is 495 and the 1991-2020 average is 771. An iceberg count above 1500 has been observed in some years, including in 2014, 2019, and between the early-1980s and mid-1990s. The all-time record of 2202 was reached in 1984. Only 2 years (1966 and 2006) in the 121-year time series reported no icebergs south of 48°N. Years with low iceberg numbers on the Grand Banks generally correspond to higher than normal air temperatures, lighter than normal sea-ice conditions, and warmer than normal ocean temperatures on the NL Shelf. The normalized anomalies of this time series is provided under the form of a scorecard below the main panel of Figure 5. This time series, starting from 1951, corresponds to the icebergs contribution to the NLCI.

## 2.5 Sea Surface Temperature

Sea Surface Temperatures (SSTs) used here are a blend of data from Pathfinder version 5.3 (1982-2020), Maurice Lamontagne Institute (1985-2013) and Bedford Institute of Oceanography (1997-2020). Monthly anomalies are computed as the average of available daily anomalies at the pixel level within each geographical region, chosen to be the NAFO Divisions of Figure 1, except that here they are cropped at the shelf break. Details of the processing are in Galbraith et al. (2020) with the extended spatial coverage as in DFO (2020a).

Figure 6 presents annual normalized anomalies averaged over the ice-free season, where the contribution of each region is weighted according to its open water area. The ice-free season varies from as short as June to September in NAFO Division 2G, to as long as March to November in NAFO Division 3P (seasonal information in the legend of Figure 6). This time series correspond to the SST contribution to the NLCI (normalized anomalies are color-coded at the bottom of the figure). This figure shows the colder than average conditions that prevailed in the early 1990's, with 1991 and 1992 being the coldest years of this time series. This period was followed by a predominance of warmer than average conditions that lasted until about 2014. In recent years, the period 2015-2019 (except for 2016) was colder than normal (defined as $\tilde{\mathrm{SST}} < -0.5\,\mathrm{sd}$; or blue colors at the bottom of Figure 6). The year 2020 was however back to above normal SST for the first time since 2014.

## 2.6 Station 27 data

Station 27 (47° 32.8'N; 52° 35.2'W) is located in the Avalon Channel just outside St. John's harbour, NL (Figure 1). It is one of longest hydrographic time series in Canada with frequent (near-monthly basis) conductivity-temperature-depth (CTD) observations since 1946. Station 27 was integrated into DFO's AZMP in 1999. In addition to sampling during these traditional

hydrographic surveys, this station has been seasonally equipped with an automatic CTD profiling system installed on a surface buoy (type Viking) since 2017 (see Cyr et al., 2020, for further information).

Station occupations were first combined into monthly averaged temperature ($T$) and salinity ($S$) profiles from which the climatological annual cycle was extracted (Figure 7). This figure shows the seasonal warming of the top layer ($\sim$20 m), with temperature peaking in August before being mixed during the fall (top panel). Also visible in the temperature field is the cold intermediate layer (CIL), a prominent feature of the NL ecosystem (Petrie et al., 1988). The CIL is defined here as the water below 0°C and delineated with thick black contour in the top panel of Figure 7. This layer originates in the winter as a cold surface layer, which becomes isolated from the surface after the apparition of a seasonally heated surface layer during the spring (April-May). The CIL thus remains below the surface throughout most of the year, while its top boundary slowly deepens from about 50 to 100 m as the heat from the surface layer penetrates deeper into the water column. While in deeper areas of the NL shelf a third warmer layer is present beneath the CIL, at Station 27 the CIL generally extend down to the bottom ($\sim$176 m). The summer CIL core temperature is defined as the minimum temperature of the monthly-averaged profile for June, July and August (see below).

The surface salinity at Station 27 is generally lowest ($S < 31$) between early-September and mid-October (Figure 7, bottom panel). These low near-surface salinities, generally from early summer to late fall, are prominent features of the salinity cycle on the Newfoundland Shelf and are largely due to the melting of coastal sea-ice upstream and carried over by the Labrador coastal current. Below the surface, the salinity at any depth increases in the spring and peaks in the summer before decreasing in the fall as a consequence of vertical mixing.

The data from Station 27, including the recent greater coverage obtained from Viking buoy automatic casts, contribute to three subindices of the NL climate index: vertically-averaged $T$ and $S$, and the CIL core temperature (Figure 8). The focus here is on the period starting in 1951 for which the seasonal coverage includes at least 8 months per year. Averages from 1980 and 1981 are excluded because of insufficient seasonal coverage (4 and 7 months, respectively). Data from 2020 have also been excluded because there were no station occupations during the first 6 months of the year due to the Covid-19 pandemic (first occupation occurred on July 14[th], the latest start since 1946). In order to account for possible changes in seasonal coverage, the annual anomalies have been calculated as the average of monthly normalized anomalies.

The vertically-averaged (0-176 m) temperature exhibits decadal-like cycles (top panel). The period between the mid-1980's and mid-1990's struck as the coldest decade of the last 70 years. It was followed by a gradual warming trend that lasted two decades and peaked in 2011, the warmest year of this time series. The period from the mid-1960's and the early-1970's was also marked by sustained warmer than normal temperatures. The end of this period coincided with the freshest anomaly on record at Station 27 observed in 1970 (center panel), an event coinciding with the Great Salinity Anomaly in the North Atlantic (Dickson et al., 1988). Similarly, the recent warmer than normal period (2010-2013) was followed by the second freshest anomaly on record observed in 2018. This most recent fresh anomaly on the NL shelf coincides with a large scale salinity anomaly in the subpolar gyre (Holliday et al., 2020), but no clear causal relationship has been established between the two. The saltiest anomaly at Station 27 (1990) occurred during the cold period of the late-1980's and early 1990's.

The CIL subindex of Station 27 is also presented in Figure 8 (bottom panel). This time series is the average normalized anomalies of the summer (June-August) CIL core temperature (minimum temperature of the monthly mean profile). The striking feature in this figure is the anomalously warm CIL anomaly present from the early 1960's to the mid-1970's. After the prevalence of a warm CIL in the early 2010's (with 2010 and 2011 being the warmest years since the 1970's), there has been a recent period of return to near normal conditions (roughly 2014-2017) that receded in 2018 and 2019. The CIL subindex of Station 27 has not been calculated for 2020.

## 2.7 Cold Intermediate Layer on the NL Shelf

As mentioned above, the CIL is a prominent feature of the NL shelf. It is found almost everywhere in subsurface during the summer. In order to highlight the influence of the CIL on the shelf as a whole, a proxy for its volume is established using the area (in $km^2$) of water below $0°C$ along Seal Island (SI), Bonavista Bay (BB) and Flemish Cap (FC) hydrographic sections (Figure 1). These sections were selected because they have been systematically surveyed since the early 1950's before being formerly standardized by the International Commission for the Northwest Atlantic Fisheries in 1976 (ICNAF, 1978). Since 1999, these hydrographic sections have been monitored by DFO as part of the AZMP.

Figure 9 shows the summer temperature along section SI during two extreme years, the warm 1965 and the cold 1990. The 1991-2020 climatology is also presented. For each sampled summer, the CIL area (e.g. the area of the section delimited by the thick black contour in Figure 9) was calculated. For example, in 1990, the area of the CIL was $26.9\,km^2$, while it was only $1.5\,km^2$ in 1965. This shows the amplitude of the interannual variability of the CIL and its potential influence on the ecosystem. In 1990, most of the sea floor along section SI was in direct contact with the CIL and water below -1°C. In 1965, when the CIL was small and fragmented, none of the sea floor was in contact with the CIL, and the bottom conditions were in consequence several degrees warmer than in 1990.

The normalized anomalies of the CIL area for sections SI, BB and FC are presented in Figure 10. This figure highlights again the warmer conditions (negative anomalies) of the 1960's and the colder conditions of the mid-1980's and early 1990's. The average of the normalized anomalies are shown in a scorecard at the bottom of the main panel and correspond to the CIL area contribution to the NLCI.

## 2.8 Bottom Temperature

Canada has been conducting random stratified trawl surveys in NAFO sub-areas 2 and 3 of the NL shelf since 1971 (Doubleday, 1981). Since 1980, temperature (and salinity since 1990) are available for most of these fishing sets thanks to trawl-mounted CTDs. The scientific trawl surveys target NAFO Subdivision 3Ps (south coast of Newfoundland) and Divisions 3LNO (Grand Banks) during the spring surveys, and Divisions 2H (northern Labrador), 2J (southern Labrador), 3K (eastern Newfoundland) and 3LNO (Grand Banks) during the fall (see map Figure 1). These surveys, combined with other available data from multiple sources (see below), are used to provide large spatial-scale oceanographic information of the NL shelf, including information on the bottom habitat parameters of numerous commercial species (e.g. Cyr et al., 2020).

The method used to derive the bottom temperature was introduced by Cyr et al. (2019) and briefly summarized here. First, all available annual profiles of temperature (scientific trawl surveys, AZMP hydrographic campaigns, surveys from other DFO regions, international oceanographic campaigns, expendable bathy-thermographs, Argo program, etc.) are vertically averaged in 5m bins and vertically interpolated to fill missing bins. Then, for each season (April-June for spring and September-December for fall), all data are averaged on a regular 0.1° x 0.1° (latitudinal x longitudinal) grid to obtain one seasonal profile per grid cell. Since this grid has missing data in many cells, each depth level is horizontally linearly interpolated. For each grid point deeper than 10 m, the bottom observation is considered as the data at the closest depth to the GEBCO 2014 Grid bathymetry (version 20141103), to a maximum 50 m difference. In order to only focus on the shelf, observations deeper than 1000 m are clipped. This method is applied for all years between 1980 and 2020 from which the 1991-2020 climatology is also derived. In order to match the scientific surveys schedule, the normalized anomalies of bottom temperature are calculated separately for NAFO Divisions 3Ps and 3LNO in the spring, and 2H, 2J, 3K and 3LNO in the fall. The time series of the bottom temperature for both seasons is presented in a stacked bar plot in Figure 11. This figure shows the cold phase from the mid-1980's to the mid-1990's, followed by a warmer phase that peaked in 2011. A scorecard at the bottom of this figure presents the mean normalized anomalies. The latter corresponds to the bottom temperature contribution to the NLCI.

## 3  Discussion

Figure 12 presents the NLCI. In the top panel, the 10 subindices described in the previous section are color-coded according to their normalized anomaly: $< 0.5$ sd in blue, $> 0.5$ sd in red, and everything within $\pm 0.5$ sd in white. Note that for some indices where positive anomalies generally indicate colder conditions (e.g., sea ice), the natural sign of the components have been reversed such that a positive value now corresponds to warm conditions (see figure caption). The subindices in their natural signs are also provided in the dataset. The middle panel of Figure 12 shows a stacked bar plot where annual NLCI values are represented by the total length of the bar (arithmetic average of all subindices available for a certain year), while individual colors have been adjusted to represent the relative contribution of each subindex to the NLCI. Numerical values of the NLCI are also reported in single a scorecard at the bottom the figure.

The NLCI highlights the different regimes prevailing on the NL shelf and the Northwest Atlantic since 1951. For example, the 1960's stands out as the warmest decade of the entire 1951-2020 period, although it is heavily driven by CIL anomalies. The following few decades have been gradually cooling until the early 1990's, with 1991 being the coldest year on records since 1951. The warming trend that followed the early 1990's peaked in 2010-2011 (depending on the record) and was followed by recent cooling that culminated in 2015. This recently observed cold period on the NL shelf (roughly 2014-2017) was the coldest period since the early 1990's (Cyr et al., 2020) and coincided with the intensification of convection in the Labrador Sea that created the largest volume of Labrador Sea Water since the early 1990's (Yashayaev and Loder, 2017).

The correlations among the different subindices are presented in Figure 13. This shows the interactions between the different components of the NLCI, while giving insights on the functioning of the NL shelf climate. The vertically-averaged temperature at Station 27 is well correlated with the bottom temperature on the NL shelf ($r = 0.80$) and with the core temperature of the CIL

($r = 0.70$). The temperature at Station 27 is also negatively correlated with sea ice ($r = -0.75$). This is because the harshness of the winter has a direct consequence on the sea ice production and the production of the CIL water. Because the CIL is in direct contact with the sea floor on a large portion of the NL shelf, it has a direct influence on its bottom temperature. It is thus not surprising that the bottom temperature of the NL shelf is also negatively correlated with sea ice ($r = -0.82$). Finally, because the air temperature has a direct influence on both SST and sea ice, the correlation is good ($r = 0.73$ and $r = -0.68$, respectively) with these variables, even if the annual air temperature average is used (i.e., includes both winter and summer). Interestingly, the winter NAO is not well correlated with any of the variables ($r < 0.5$), except slightly with the number of icebergs ($r = +0.52$). While the winter NAO is expected to capture some decadal dynamics (warm 1960's and early 2010's; cold 1990's and late 2010's), the weak correlation between the NAO and the different NLCI components suggests that lag or inertial effects are important on the NL shelf (e.g. it takes several consecutive cold winters to build an important volume of cold water). It also suggests that the NLCI captures other types of variability specific to the NL shelf compared to the winter NAO alone. The good correlation between Station 27 temperature and other components of the climate index adds weight to numerous studies suggesting that this station is representative of the large-scale climate of the NL shelf (e.g. Petrie et al., 1991, 1992; Colbourne et al., 1994; Drinkwater, 1996; Han et al., 2015).

Figure 13 also includes the correlation coefficients between the NLCI index and all its subindices. Although a visual examination of Figure 8 shows matching warm/fresh and cold/salty periods at Station 27, salinity is not significantly correlated with any of the subindices of the NLCI. This suggests that the relation between freshwater and water temperature do not operate on a year to year basis, but may lagged or relevant on longer (e.g. decadal) time scales. Salinity was however kept in the NLCI because it has recently been shown to be a useful predictor for the NL ecosystem (e.g. submitted study on capelin spawning dynamics), and because freshwater fluxes on the NL shelf are an important contribution to the freshwater budget of the North Atlantic, with consequences on the North Atlantic overturning circulation (Florindo-López et al., 2020). Because the 10 subindices are provided here, users of the NLCI can however recompute their own climate index using any combination of the subindices. Except for salinity at Station 27, the correlation coefficient between the NLCI and its subindices varies between $|r| = 0.58$ and $|r| = 0.89$, the latter with the NL shelf bottom temperature. The absence of very strong correlation (e.g. $|r| > 0.90$) between the different subindices, and between any subindex and the climate index itself, shows the relative independence of the different components of the climate index, and gives confidence in the choices made for its design.

Finally, there is a good correlation ($r = 0.87$) between this new climate index and the averaged version of the CEI used until recently (e.g., Cyr et al., 2019), ensuring continuity with previous studies using this index (Figure 14). A notable difference between the two indices, however, is that the variance of the former ($\sigma^2 = 1.2$) is larger than that of the latter ($\sigma^2 = 0.4$; note the different vertical axis systems in Figure 14). It is likely that the greater independence of the components used here reduces the variance of the new climate index compared with the previous CEI that used 28 components, some being highly correlated (e.g., air temperature at 4 sites, SST in overlapping or close areas, bottom temperature and salinity in neighbouring regions, etc.).

## 4 Conclusions

This article describes a new climate index for the Newfoundland and Labrador shelf. This index is composed of 10 subindices representing different aspect of the NL ecosystem: winter NAO, air temperature, sea ice season severity, icebergs count, SST, Station 27 temperature, salinity and CIL core temperature, CIL area on 3 hydrographic sections, and bottom temperature on

the NL shelf. Some subindices are season specific (e.g. winter NAO, icebergs and sea ice season severity), while others are representative of the entire annual cycle (e.g., air temperature and seasonal SST), which may mask strong seasonal contrasts during some years. Because the NLCI and all 10 subindices are made available, users can derive their own custom index by averaging any combination of subindices. It is expected that this new index will be useful for ecosystem studies, stock assessments, forecast models of marine resources, and more.

## 5 Data availability

The data presented in this study are available here: https://doi.org/10.20383/101.0301 (Cyr and Galbraith, 2020). Three comma-separated values (CSV) files are provided:

– *NL_climate_index.csv*: Annual values of the NL climate index.

– *NL_climate_index_all_fields.csv*: Annual values of the 10 subindices making the NL climate indices (with some signs

reversed, see Figure 12 caption). The average of these 10 subindices correspond to the NL climate index.

– *NL_climate_index_all_fields_natural_signs.csv*: annual values of the 10 subindices making the NL climate indices in with their natural sign.

For the foreseeable future, this index will be updated on an annual basis once the update from the previous year is completed (e.g., targeted release in the spring of the following year).

*Author contributions.* F. Cyr designed the study, lead the writing and calculated most of the subindices. P.S. Galbraith calculated the SST and the sea ice subindices, and participated in the writing. Both authors discussed in depth the science, technical details and the structure of this article.

*Competing interests.* The authors declare that no competing interests are present

*Acknowledgements.* This work is a contribution to the Atlantic Zone Monitoring Program (AZMP) of Fisheries and Oceans Canada (DFO).
The authors thank the numerous scientists, technicians, captains and crew members who participated to the sampling and analysis effort

since 1951. The authors also thank Dr. Pierre Pepin, who provided comments on an early version of the manuscript and Dr. Bee Berx and two anonymous reviewers for their valuable comments on the study.

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

**Table 1.** Review of AZMP reports that made use of composite or climate indices. The Table is separated into three sections: 1) Meteorological, sea ice and SST conditions off eastern Canada; 2) Physical oceanographic conditions (review year 2005-2008) and meteorological and sea ice conditions (after 2009) on the Scotian shelf and Gulf of Maine; and 3) Physical oceanographic conditions (including sea ice and atmosphere) on the and Newfoundland and Labrador shelf. Note that the entries between Hebert et al. (2011) Hebert et al. (2020) have not been included because their ocean composite index did not change. The different columns are, respectively, the year in review, the number of components in the composite index, the time series used and the name given to this index in the document.

**Eastern Canada (Ocean and atmosphere index)**

| | Year | No. comp. | Types of components[*] | Name |
|---|---|---|---|---|
| **Petrie et al. (2006a)** | 2005 | 23 | NAO, airT, ice, SST | N/A |
| **Petrie et al. (2007a)** | 2006 | 23 | NAO, airT, ice, SST | Composite index |
| **Petrie et al. (2008a)** | 2007 | 23 | NAO, airT, ice, SST | Composite index |
| **Petrie et al. (2009a)** | 2008 | 23 | NAO, airT, ice, SST | Composite index |

**Scotian shelf and Gulf of Maine (Ocean only index)**

| | | | | |
|---|---|---|---|---|
| **Petrie et al. (2006b)** | 2005 | 18 | botT, stationT | N/A |
| **Petrie et al. (2007b)** | 2006 | 18 | botT, stationT | Composite index |
| **Petrie et al. (2008b)** | 2007 | 18 | botT, stationT | Composite index |
| **Petrie et al. (2009b)** | 2008 | 18 | botT, stationT | Composite index |
| **Hebert et al. (2011)** | 2009-10 | 18 | botT, stationT | Composite index |
| **...** | ... | ... | ... | ... |
| **Hebert et al. (2020)** | 2018 | 18 | botT, stationT | Composite index |

**Newfoundland and Labrador shelf (Ocean and atmosphere index)**

| | | | | |
|---|---|---|---|---|
| **Colbourne et al. (2006)** | 2005 | 0 | N/A | N/A |
| **Colbourne et al. (2007)** | 2006 | 0 | N/A | N/A |
| **Colbourne et al. (2008)** | 2007 | 26 | NAO, airT, ice, icebergs, SST, S27, botT, CIL, secT | Composite index |
| **Colbourne et al. (2009)** | 2008 | 26 | NAO, airT, ice, icebergs, SST, S27, botT, CIL, secT | Composite climate index |
| **Colbourne et al. (2011)** | 2010 | 26 | NAO, airT, ice, icebergs, SST, S27, botT, CIL, secT | Composite climate index |
| **Colbourne et al. (2012)** | 2011 | 26 | NAO, airT, ice, icebergs, SST, S27, botT, CIL, secT | Composite climate index |
| **Colbourne et al. (2013)** | 2012 | 26 | NAO, airT, ice, icebergs, SST, S27, botT, CIL, secT | Composite climate index |
| **Colbourne et al. (2014)** | 2013 | 26 | NAO, airT, ice, icebergs, SST, S27, botT, CIL, secT | Composite climate index |
| **Colbourne et al. (2015)** | 2014 | 28 | NAO, airT, ice, icebergs, SST, S27, botT, CIL, secT | Composite climate index |
| **Colbourne et al. (2016)** | 2015 | 28 | NAO, airT, ice, icebergs, SST, S27, botT, CIL, secT | Composite climate index |
| **Colbourne et al. (2017)** | 2016 | 28 | NAO, airT, ice, icebergs, SST, S27, botT, CIL, secT | Composite climate index |
| **Cyr et al. (2019)** | 2017 | 28 | NAO, airT, ice, icebergs, SST, S27, botT, CIL, secT | Composite climate index |
| **Cyr et al. (2020)** | 2018 | 10 | NAO, airT, ice, icebergs, SST, S27, botT, CIL | NL climate index |

[*] *NAO* = winter NAO; *airT* = air temperature at different sites; *ice* = sea ice parameters; *SST* = Sea Surface Temperature in different sub-areas; *botT* = bottom temperature in different sub-areas; *stationT* = temperature at monitoring stations (different depth ranges); *S27* = Station 27 temperature and salinity (different depth ranges); *CIL* = cold intnermediate layer parameters; *secT* = mean temperature along repeated hydrographic sections

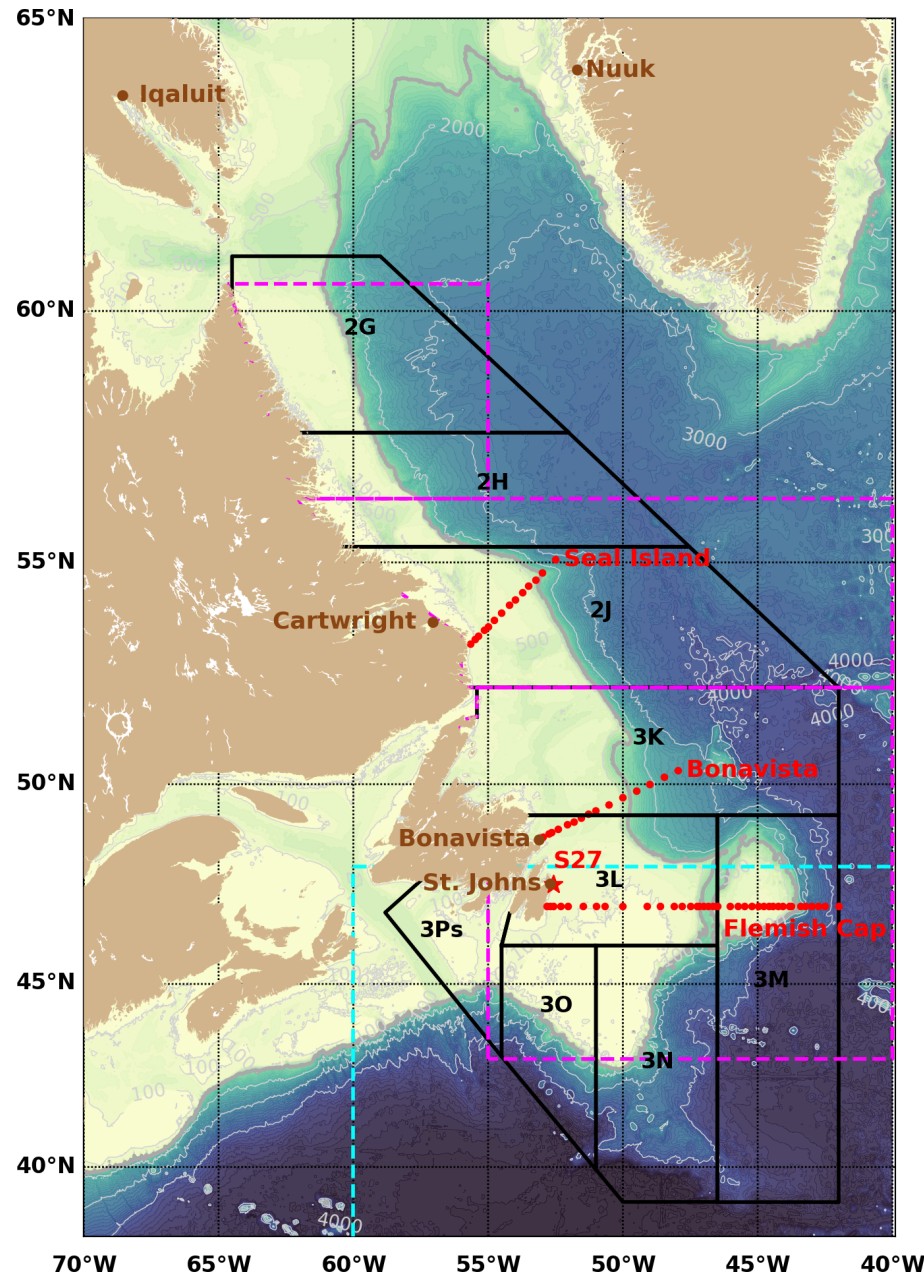

**Figure 1.** Map and main bathymetric features of the Northwest Atlantic ocean. NAFO Divisions (sub-areas 2 and 3) on the Newfoundland and Labrador (NL) shelf are drawn. The AZMP hydrographic sections Seal Island, Bonavista Bay and Flemish Cap are are shown with red dots. Long-term AZMP hydrographic Station 27 is highlighted with red star. The five stations used for the air temperature time series are shown in brown. The 3 regions used for sea ice calculations are drawn with dashed magenta lines: Northern Labrador shelf, Southern Labrador shelf and Newfoundland shelf, respectively from north to south. The region used by the IIP for iceberg sighting south of 48°N is drawn in dashed cyan. The shelf break is delimited by a thicker and darker contour corresponding to the isobath 1000 m (used to clip the SST and bottom temperature).

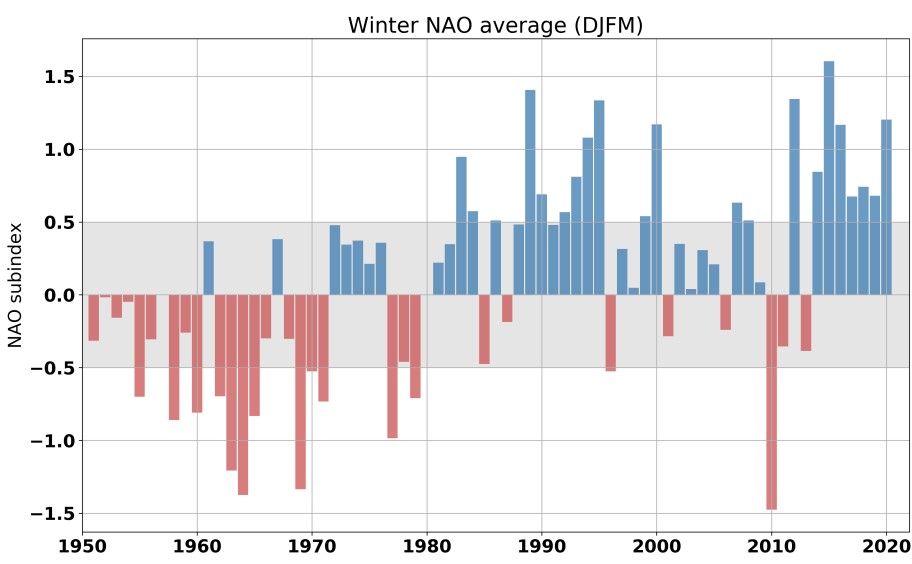

**Figure 2.** Winter North Atlantic Oscillation (NAO) index averaged over December to March. Here positive anomalies, generally indicative of colder conditions, are colored in blue. Shaded area correspond to $\pm0.5$ sd, indicating normal conditions. This time series is one component of the NL climate index.

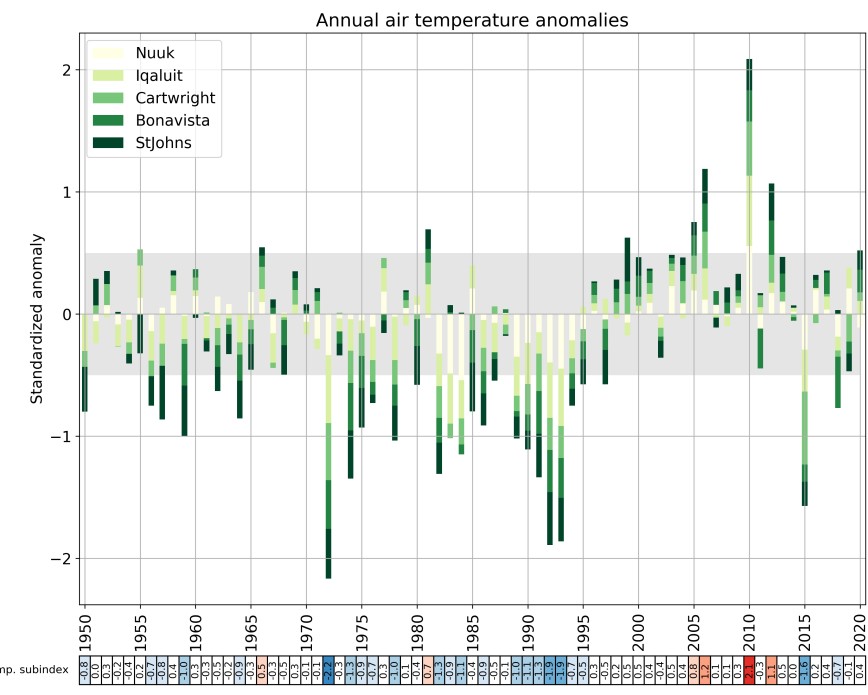

**Figure 3.** Normalized annual air temperature anomalies for Nuuk, Iqaluit, Cartwright, Bonavista and St. John's. This figure shows the average of the five stations, where the length of each bar correspond to the relative contribution of individual station to the average. The shaded area corresponds to the 1991-2020 average $\pm 0.5$ sd. The numerical values of this time series are reported in a color-coded scorecards at the bottom of the figure. Positive anomalies ($> 0.5$ sd) are colored red, while negative anomalies ($< -0.5$ sd) are colored blue. In both cases, the darker the color, the stronger the anomaly. White corresponds to the climatological average $\pm 0.5$ sd. This time series is one component of the NL climate index.

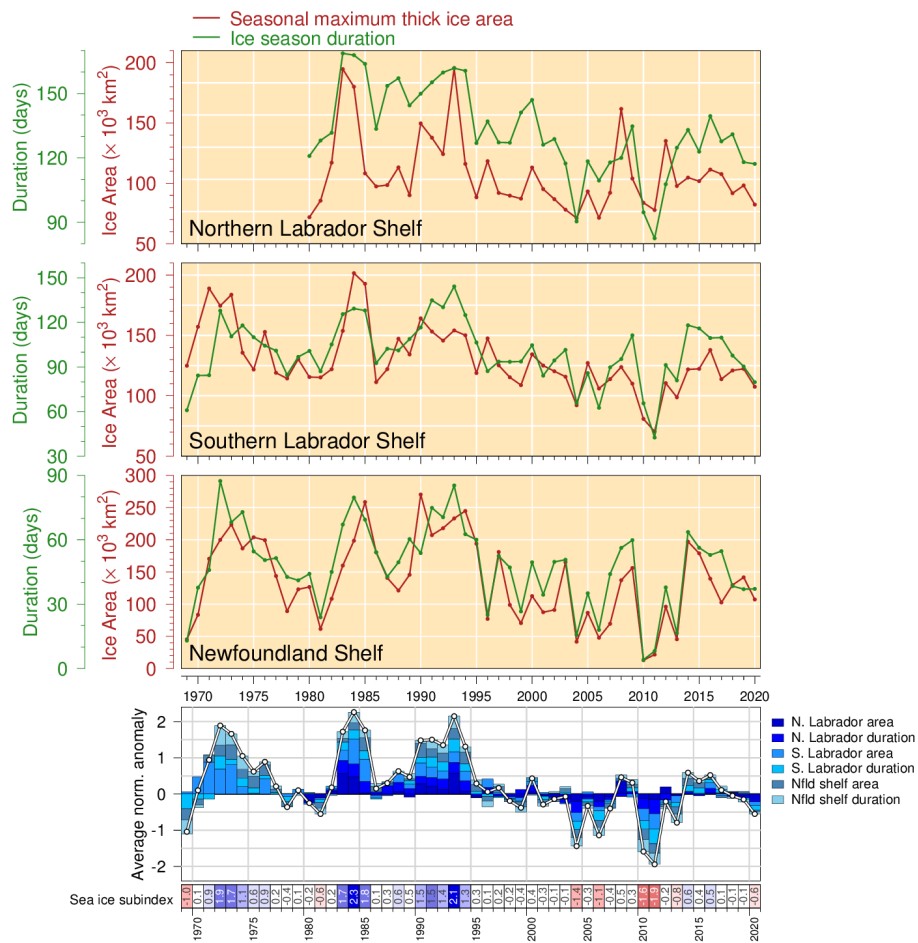

**Figure 4.** Sea ice season duration (green) and seasonal maximum area covered by thick ice (red) for Northern and Southern Labrador shelf (top two panels) and Newfoundland shelf (third panel). The Northern Labrador time series starts in 1980, while the two other start in 1969. The 6 time series are transformed into normalized anomalies and presented in a stacked bar fashion in the bottom panel. The scorecard at the bottom of this last panel presents the numerical values of the mean normalized anomalies for each year. Here negative anomalies (indicative of warmer conditions) are colored red and positive anomalies (colder conditions), blue. This time series is one component of the NL climate index.

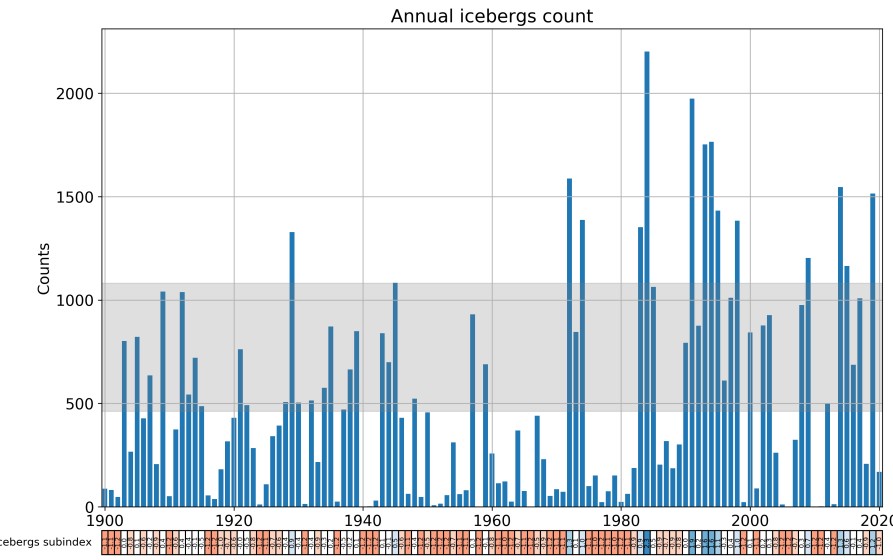

**Figure 5.** Annual iceberg count crossing south of 48°N on the northern Grand Bank. The shaded area corresponds to the 1991-2020 average ±0.5 sd. The normalized anomaly of this time series is provided below the main panel under the form of a color-coded scorecard. Here negative anomalies are red (generally corresponding to warmer conditions) and positive anomalies blue. This time series is one component of the NL climate index.

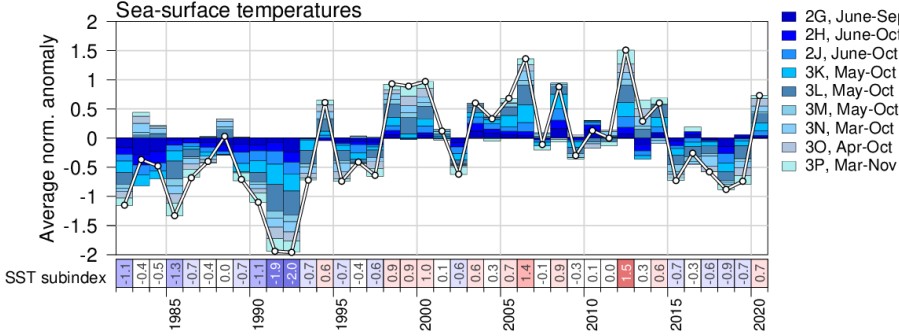

**Figure 6.** Sea Surface Temperature (SST) composite index in NAFO sub-areas 2 and 3 since 1982. This index is built by performing a spatially-weighted average of the seasonal SST normalized anomalies for each NAFO division (divisions and months used are provided on the right-and side of the figure). The numerical values of the normalized anomalies are provided in a color-coded (red, warm; blue, cold) scorecard at the bottom. This time series is one component of the NL climate index.

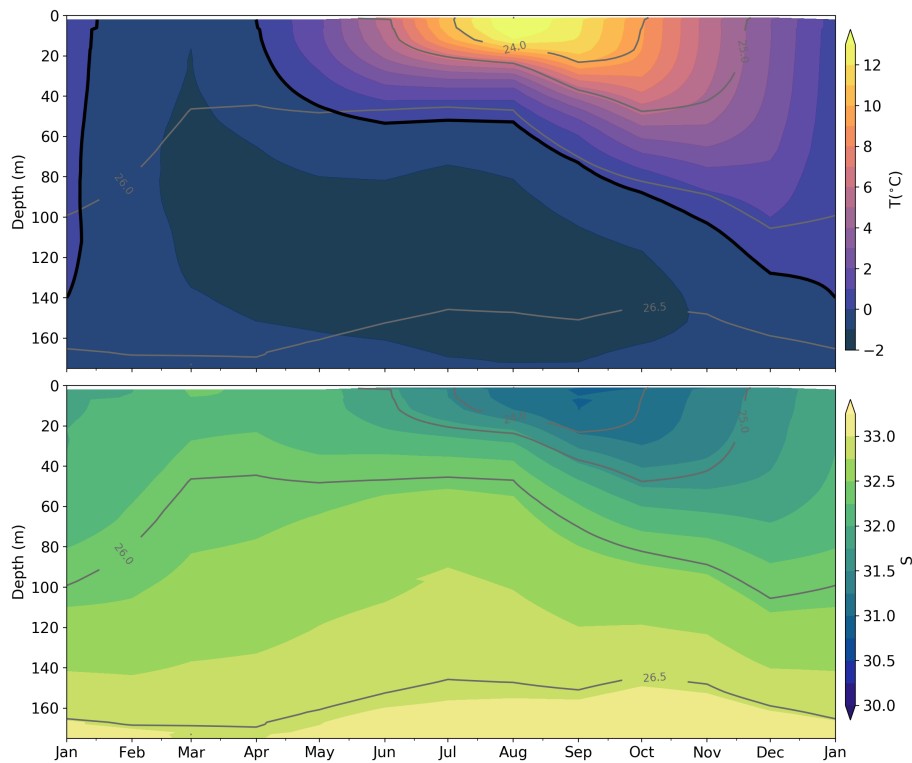

**Figure 7.** Climatological (1991-2020) annual cycle of temperature (top) and salinity (bottom) at Station 27. The gray contours are the isopycnals ($\sigma_0$ in $\mathrm{kg\,m^{-3}}$) derived from temperature and salinity using the TEOS-10 package (McDougall and Barker, 2011). The thick black line in the top panel is the $0°\mathrm{C}$ isotherm delimiting the top of the CIL.

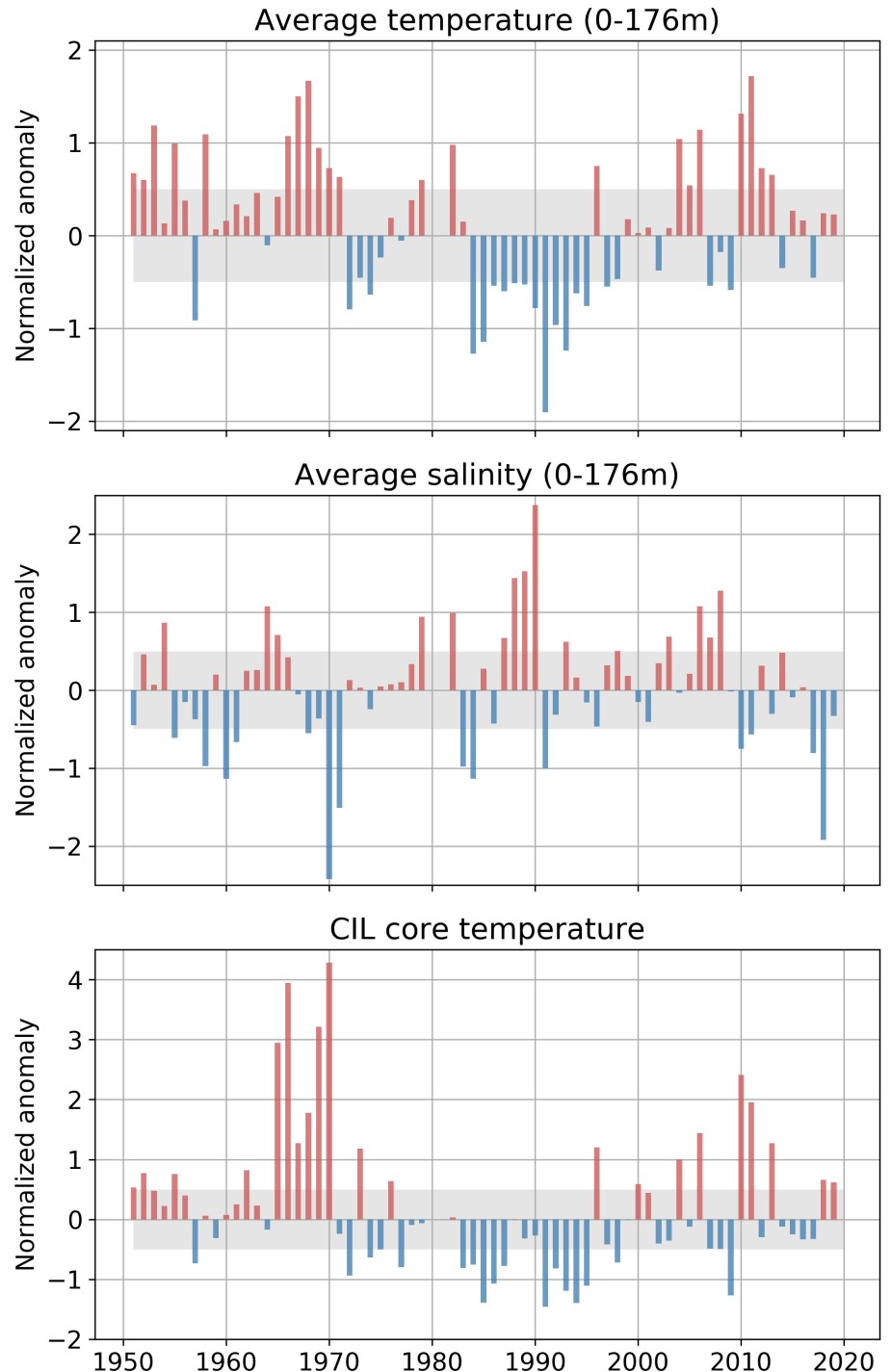

**Figure 8.** Normalized anomalies of the vertically averaged temperature (top) and salinity (center), and CIL core temperature (bottom) at Station 27. Shaded areas in all panels corresponds to the 1991-2020 average ±0.5 sd. These three time series are components of the NL climate index.

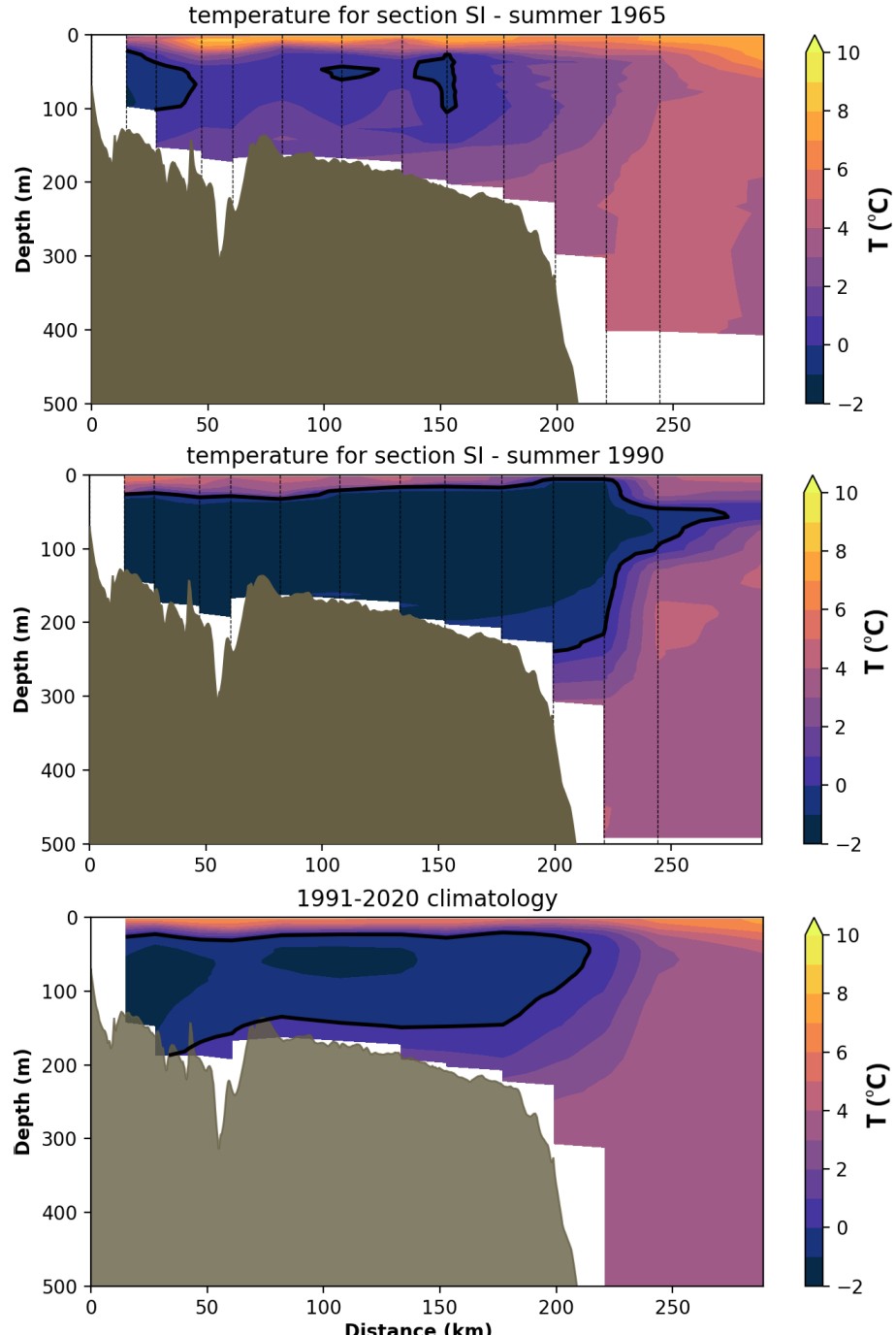

**Figure 9.** Summer temperature along hydrographic section Seal Island (SI) during 1965 (top), 1990 (center) and the 1991-2020 average (bottom). The vertical axis is limited to the top 500 m and the horizontal axis is the distance (in km) from the coast. The CIL (T<0°C) is highlighted with a thick black contour. Stations location are indicated with thin dashed lines.

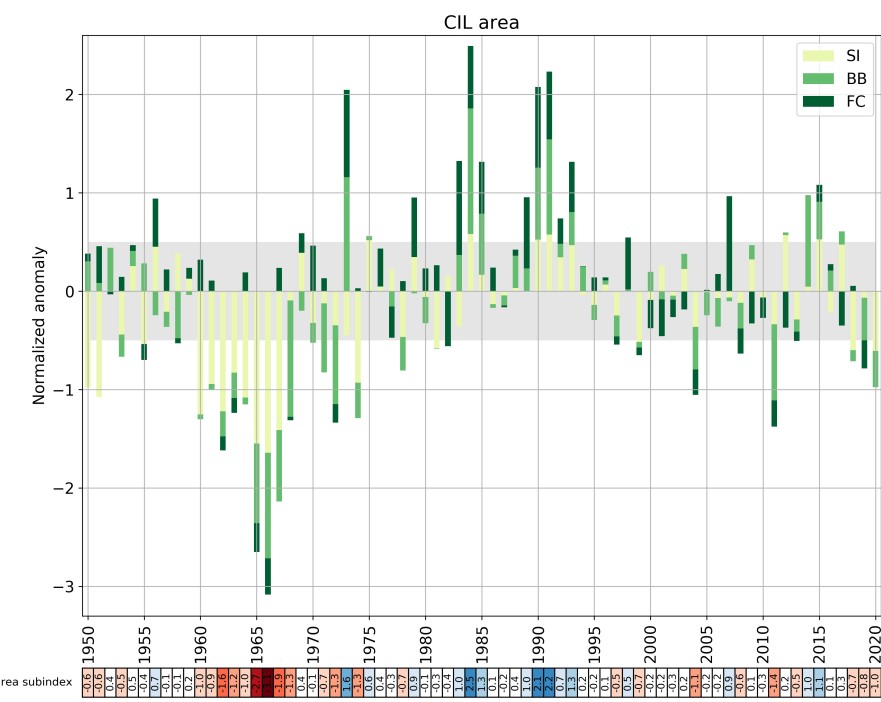

**Figure 10.** Normalized anomalies of the mean CIL area for hydrographic sections Seal Island (SI), Bonavista Bay (BB) and Flemish Cap (FC). This time series correspond to the average of the 3 sections, where the contribution of each section is represented. The shaded area corresponds to the 1991-2020 average ±0.5 sd. The numerical values of this time series are reported in a color-coded scorecards at the bottom of the figure. Here negative anomalies (generally corresponding to warmer conditions) are colored red, and positive anomalies blue. This time series is one component of the NL climate index.

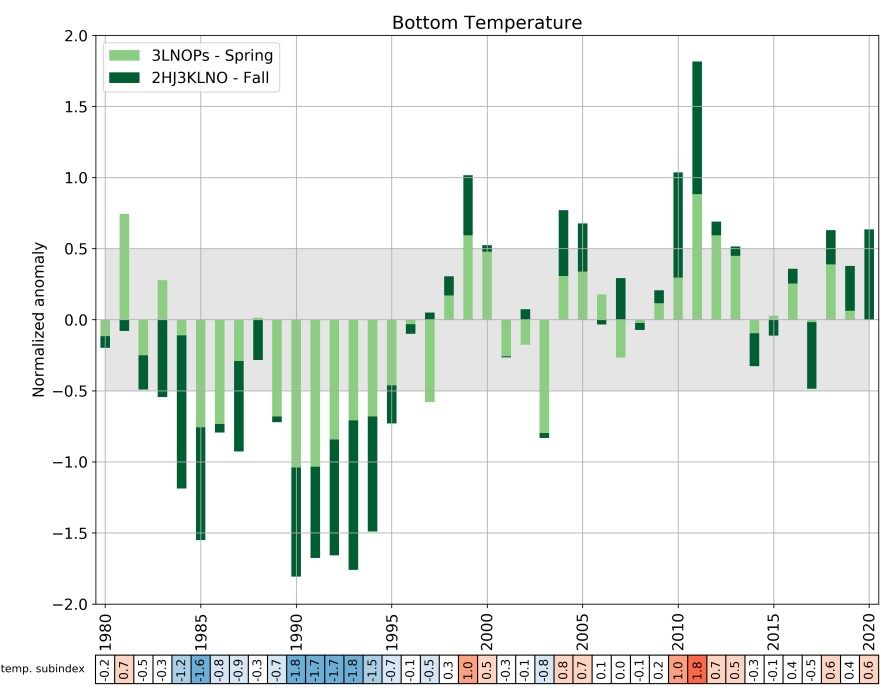

**Figure 11.** Normalized anomalies of bottom temperature in NAFO Dividions 3LNOPs (spring) and 2HJ3KLNO (fall). This time series corresponds to the average of the 2 seasons, where each contribution is represented. The shaded area corresponds to the 1991-2020 average ±0.5 sd. The numerical values of this time series are reported in a color-coded scorecards at the bottom of the figure. This time series is one component of the NL climate index.

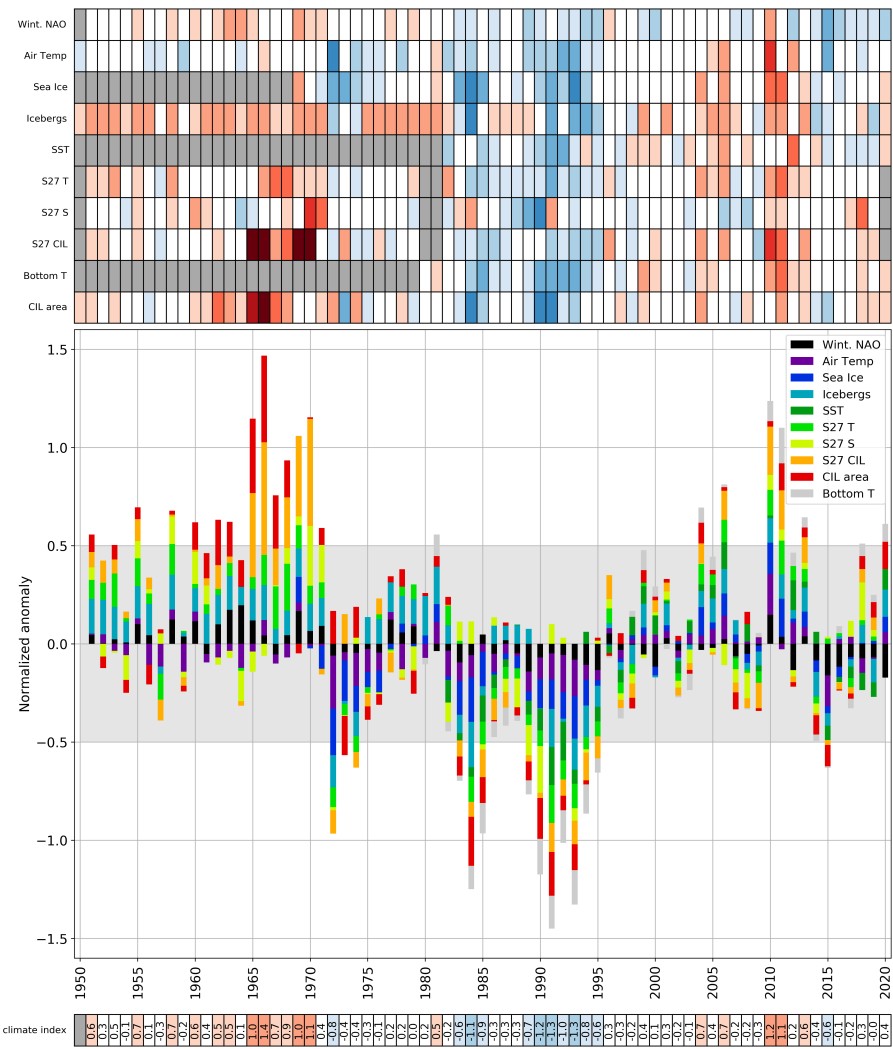

**Figure 12.** Newfoundland and Labrador climate index described in this study. The scorecard at the top of the figure represents the 10 sub-indices used to construct the climate index, color-coded according to their value (blue negative, red positive, white neutral). These time series are the following: winter NAO index (starts in 1951), the air temperature at 5 sites (starts in 1950), the sea ice season duration and maximum area for the Northern Labrador, Southern Labrador and Newfoundland shelves (starts in 1969), the number of icebergs (starts in 1950), SSTs in NAFO divisions 2GHJ3KLNOP (starts in 1982), vertically-averaged temperature and salinity at Station 27, CIL core temperature at Station 27 (starts in 1951), the summer CIL areas on the hydrographic sections Seal Island, Bonavista Bay and Flemish Cap (starts in 1950), and the spring and fall bottom temperature in NAFO divisions 3LNOPs and 2HJ3KLNO, respectively (starts in 1980). The sign of some indices (NAO, ice, icebergs, salinity and CIL volume) have been reversed when positive anomalies are generally indicative of colder conditions. Gray cells in the scorecards indicate the absence of data. The center panel of the Figure represents the climate index in a stacked-bar fashion, where the total length of the bar is the average of the respective sub-indices, and where their relative contribution to the average is adjusted proportionally. The scorecard at the bottom of the figure shows the color-coded numerical values of the climate index.

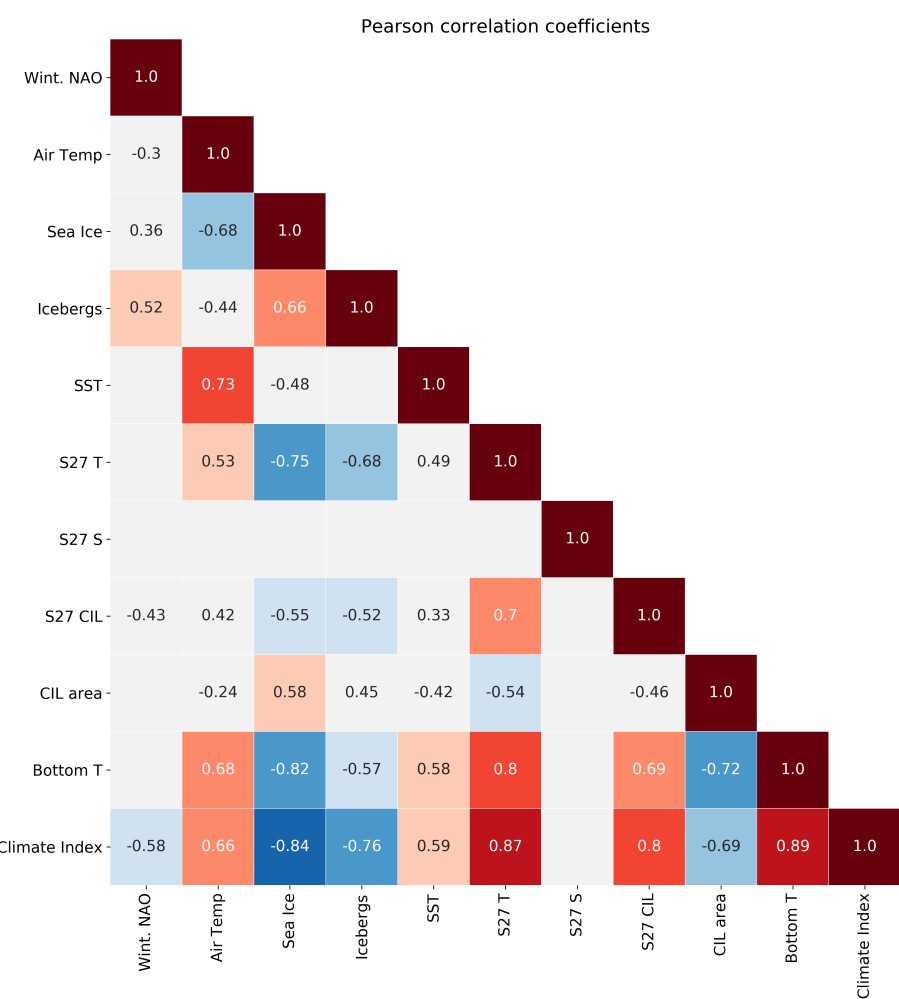

**Figure 13.** Pearson correlation ($r$) matrix between the different sub-indices of the NL climate index, and the NL climate index itself. Red and blue colors denote a positive and negative correlation, respectively. Only significant correlations (p-values $< 0.05$) are shown. Correlations less than $\pm 0.5$ have been left white. The natural signs (not reversed) of the subindices have been used here in order to illustrate relationships in which positive (warm) anomalies in one variable (e.g., Station 27 temperature) is reflected by a negative anomalies in another (e.g., sea ice).

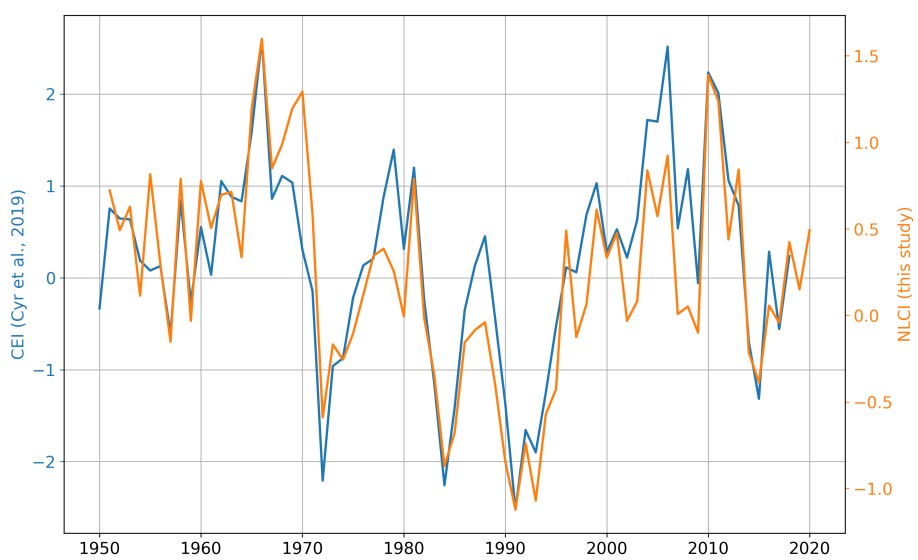

**Figure 14.** Comparison between the Composite Environmental Index (CEI) used until recently (e.g. Cyr et al., 2019), and the new NL climate index introduced here. Note the difference vertical axis systems. The correlation between the two previous and the new indices is $r = 0.87$. Note that in order to be consistent with the last version of the CEI, the NLCI based on the 1981-2010 climatology was used here. The CEI is also presented here as the averaged of the 28 component, rather than its sum.

## Appendix A: Figures using 1981-2010 climatology

The change in the climatological period from 1981-2010 to 1991-2020 is a shift towards a warmer reference period (exclusion of the cold mid-1980's and inclusion of the warmer mid-2010's). The consequences of this shift for the different subindices is generally an exacerbation of the colder anomalies and a reduction of the amplitude of the warmer anomalies. This appendix provides alternate versions of the figures provided in the manuscript that used a 1991-2020 climatology, but this time using the previous climatological period of 1981-2010.

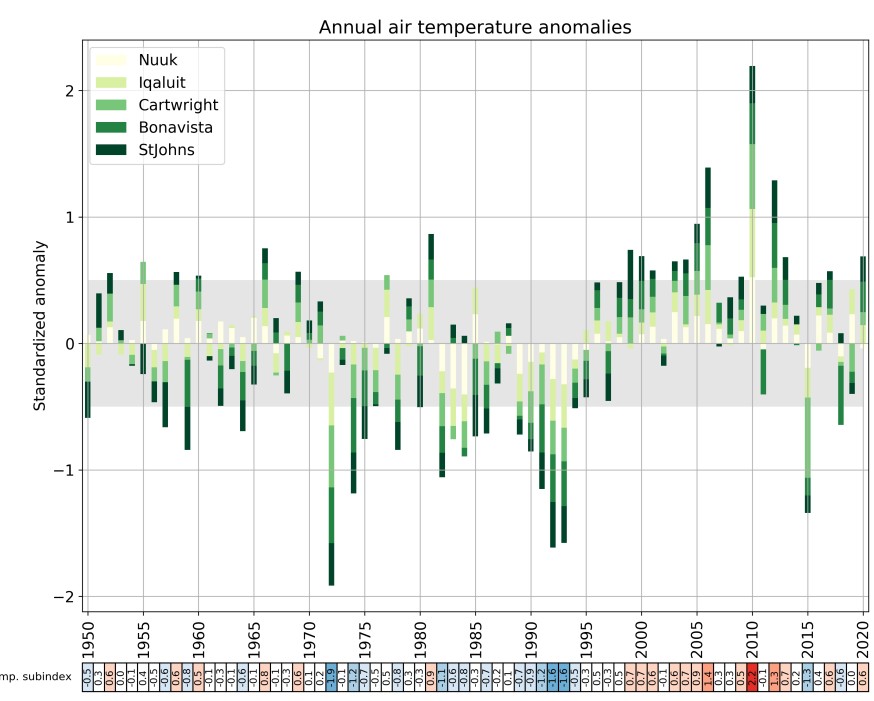

**Figure A1.** Same as Figure 3, but using a climatology referenced to the 1981-2010 period.

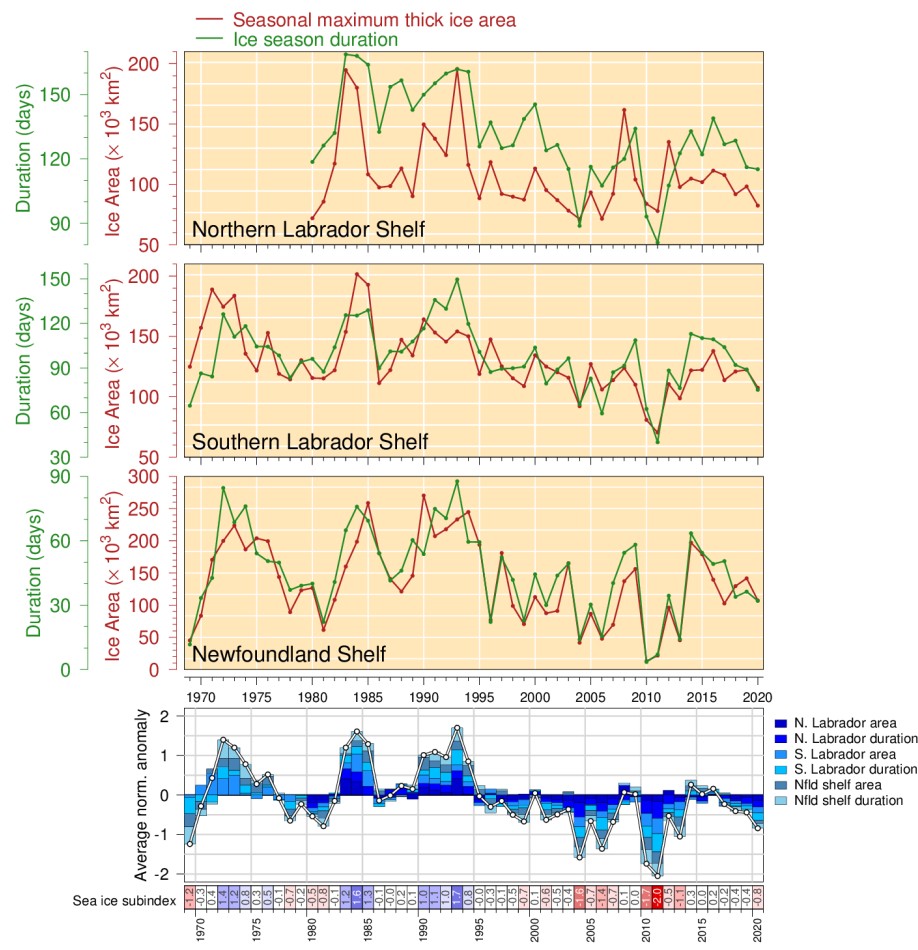

**Figure A2.** Same as Figure 4, but using a climatology referenced to the 1981-2010 period.

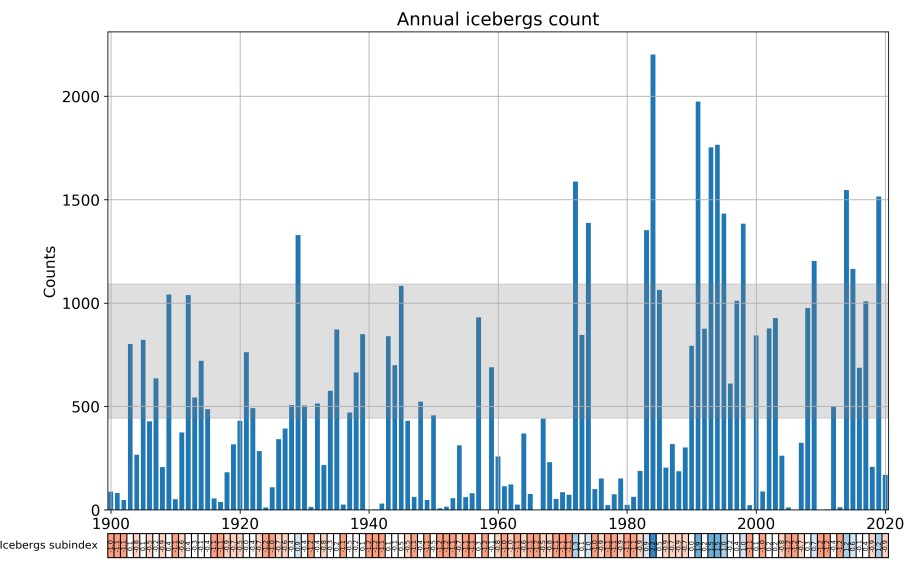

**Figure A3.** Same as Figure 5, but using a climatology referenced to the 1981-2010 period.

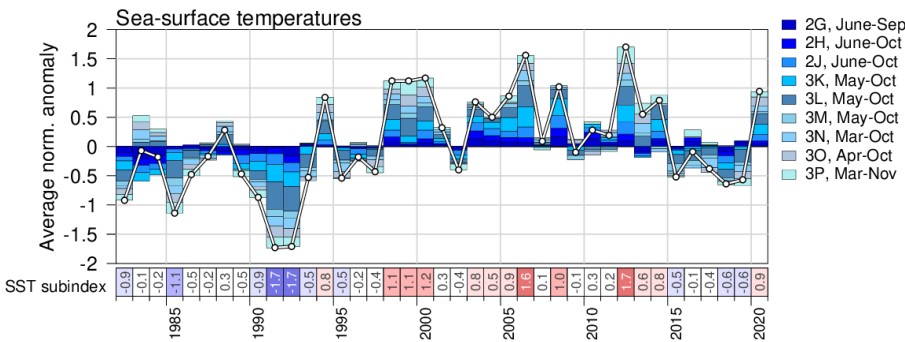

**Figure A4.** Same as Figure 6, but using a climatology referenced to the 1981-2010 period.

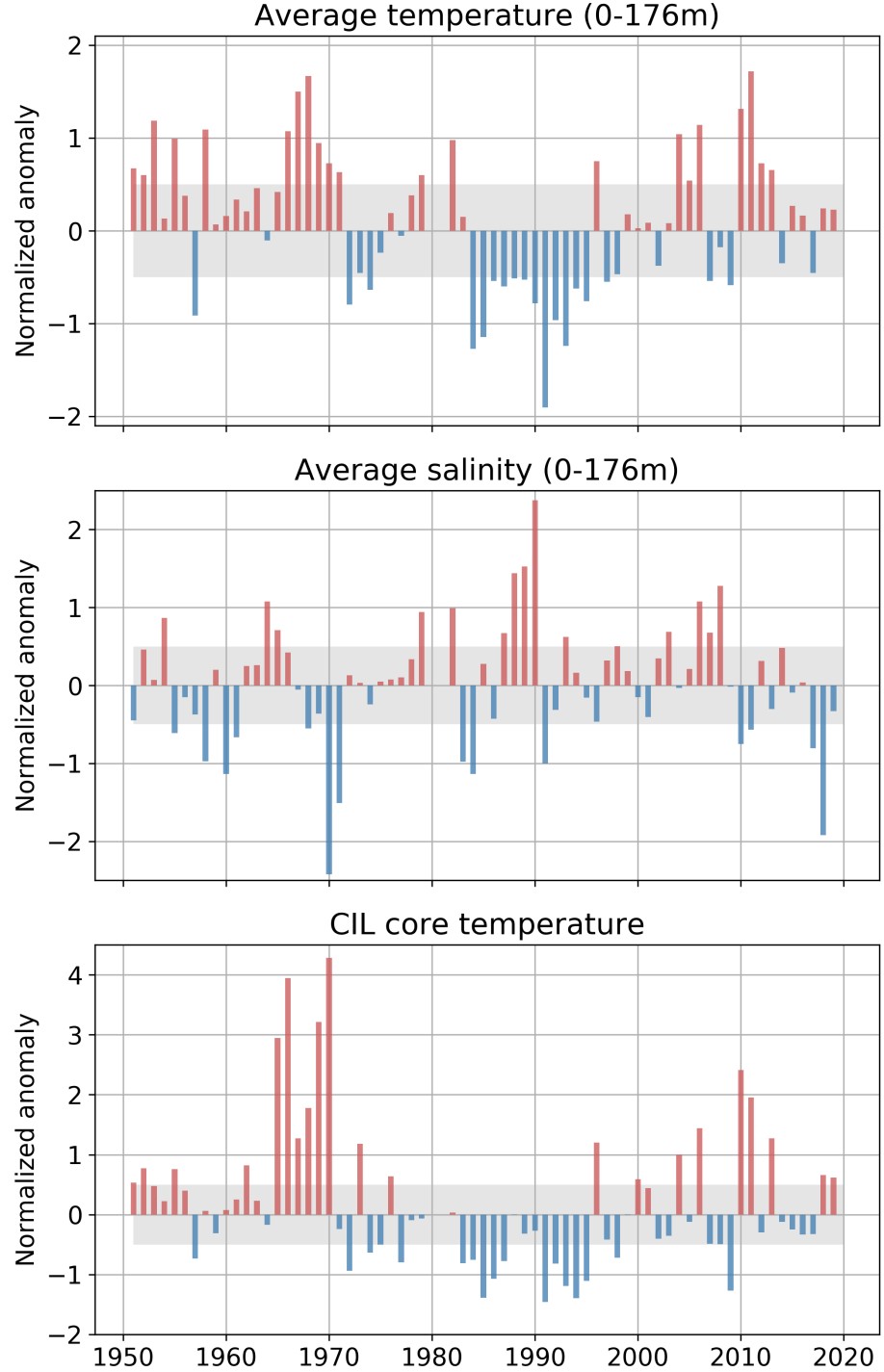

**Figure A5.** Same as Figure 8, but using a climatology referenced to the 1981-2010 period.

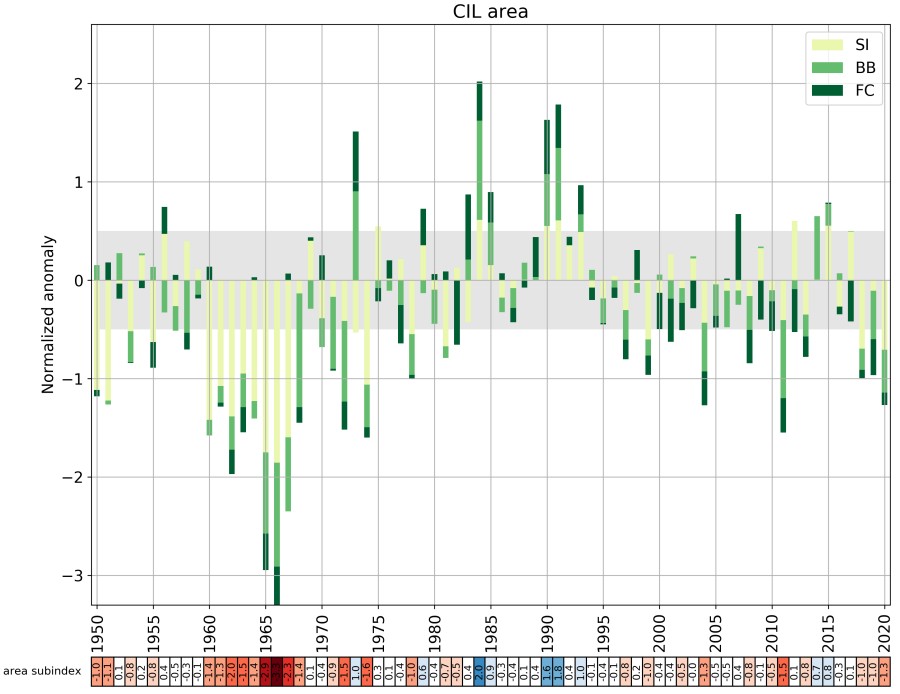

**Figure A6.** Same as Figure 10, but using a climatology referenced to the 1981-2010 period.

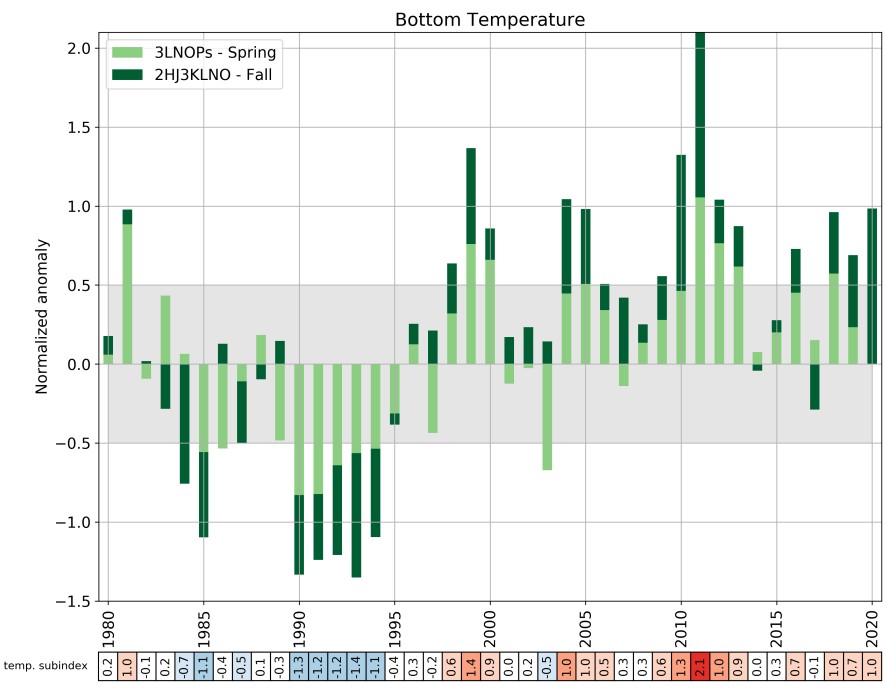

**Figure A7.** Same as Figure 11, but using a climatology referenced to the 1981-2010 period.

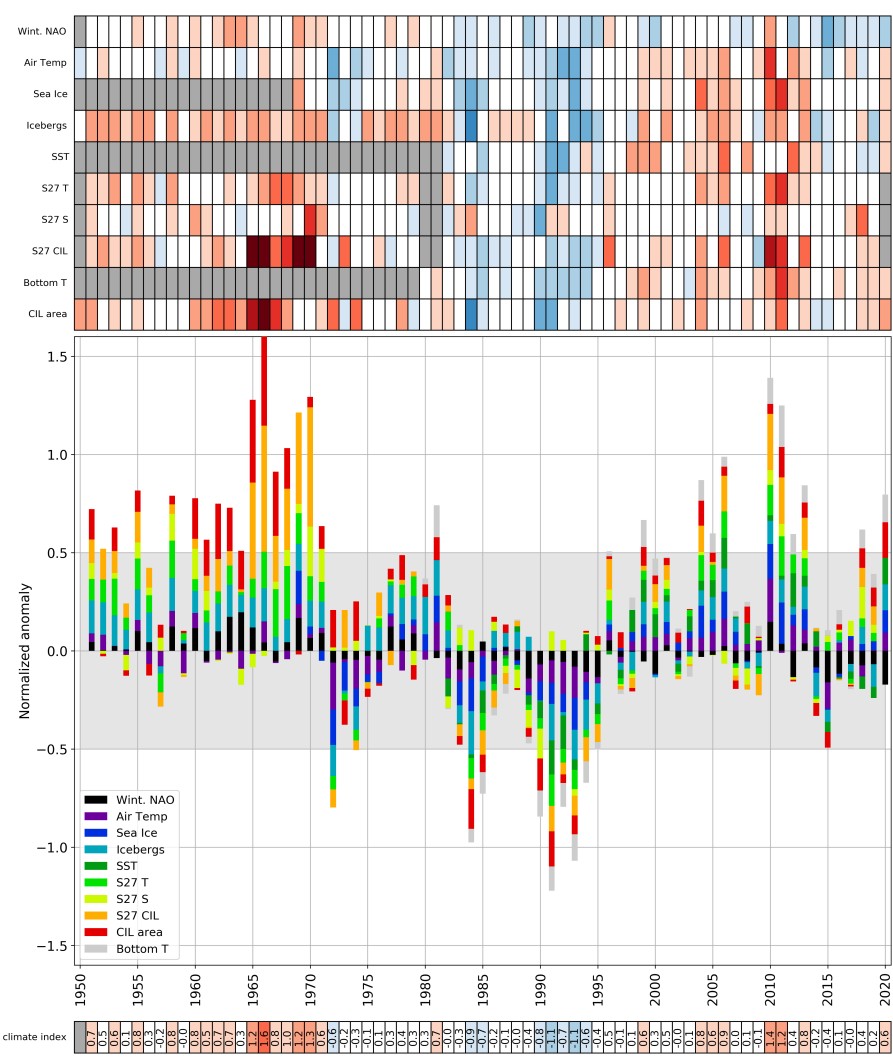

**Figure A8.** Same as Figure 12, but using a climatology referenced to the 1981-2010 period.