# Peer review of "A climate index for the Newfoundland and Labrador shelf"

_Earth System Science Data, 2020_

## Referee Comment (RC1) · Barbara Berx (Referee) · 8 Jan 2021

In "A climate index for the Newfoundland and Labrador shelf", Cyr and Galbraith present a new method to combine ten meteorological and physical oceanographic quantities into a single indicator for the shallow shelf areas adjacent to Newfoundland and Labrador, Canada. The manuscript is well written with a good structure. In a small number of instances, the language on how the indicators were calculated could be clarified further to avoid ambiguity. The provision of index time series that can be applied in marine research is very welcome, and I do appreciate the efforts the authors to translate measurements across the region into a meaningful data product. However, I did have some more significant questions on the calculations, on the index and what it means from an oceanographic perspective and its subsequent relationships through

the marine food web.

These questions/suggestions may be more significant to address, hence my choice for major revisions, but I would strongly encourage the authors to consider these and respond to the review as I can see great benefit to a comprehensive manuscript to accompany what I can see as a valuable data product.

Item 5 below, I would personally consider to merely improve the manuscript, and more importantly the application and use of the data product by others, and I would welcome the authors to make their own decision on whether or not to put in the effort to address this.

1) Calculation of some of the sub-indices: I have some questions around the method and choice of some of the sub-indices, their combination and/or inclusion.

- 2.1 NAO: The choice of the EOF-based NAO means that technically the time series will be slightly different each year (due to the nature of the analysis). Similar to the information highlighted in the Hurrell product, the users of the NLCI should be made aware that this means they do need to download the entire time series annually (rather than add a single value to the end of their time series). The authors could avoid this requirement by choosing the alternate NAO data product. If the current method is maintained, the caveat does need to be made explicit to ensure awareness with end-users.

- 2.2 Air Temperature: The inclusion of the more remote sites of Nuuk and Iqaluit needs some further explanation. How does weather at these remote sites impact ocean state on the Labrador and Newfoundland Shelf? Looking at Figure 3 (but please note my comment on the figures in point 4), it looks like the sign of anomalies at these sites is at times opposite to those more local to the Labrador and Newfoundland Shelf. The authors may want to consider giving the local and remote weather conditions separate weight in their combined index. The analysis of how the different component indices correlate may also show some stronger/clearer signals if this is done.

- 2.3 Sea Ice: The language around the calculation of combining the 6 time series (2 variables over 3 regions) is somewhat ambiguous. My understanding is that normalised anomalies were calculated and then averaged (arithmetic mean), correct?

- 2.4 Iceberg Count: The region where the measurement is made should be included in the map in Figure 1.

- 2.5 Station 27: The climatological conditions at Station 27 suggest the region stratifies to some extent throughout the year (see suggestion on isopycnals for Figure 7). The vertical average temperature and salinity may therefore be masking important variability in the near-surface and near-bed layer which may be driven by different processes. The authors also describe a three-layer system at the sampling location, which to me suggests that a vertical average is possibly not the most representative of on-shelf conditions. The complete lack of significant correlation of the S27 salinity is one indication to me that the choices here should be reconsidered. Some questions for the authors to consider are: could S27 surface and near-bed salinity and temperature be treated separately? Is one of these more/less relevant for the ecosystem of the region (for example, are there known links between fish stocks and recruitment success and one/several of these climate variables?)? Could the strength of stratification (for example expressed as potential energy anomaly) or the size of salinity/temperature range in the year be more important (on lines 115-117 the authors highlight the importance of the salinity cycle, but this is not adequately reflected in the sub-index or eventual climate index). Generally, salinity is a good indicator of circulation change, and therefore I would have suspected it to play a more important role, particularly due to the sub-polar gyre's influence on the region (see also item 3 below).

- 2.8 Bottom temperature: What is the reason for choosing the 1000 m isobath to delimit the extent of the shelf? Most publications consider the boundary to be the 200 or 500 m isobaths. Does this broader extent increase data availability? Is it because this is what the fisheries assessments use? Does this definition mean that a significant portion of water masses from deeper in the Labrador Sea is included? How does

[Figure]

this inclusion of deeper water (which is likely not influence by the same processes as the shallow shelf region) significantly impact on the bottom temperature mean and its seasonal/inter-annual variability?

2) The oceanographic understanding behind the combined index: This index will be very valuable to other marine scientists studying the ecosystem dynamics and productivity of the region. There is little interpretation of this throughout the manuscript (see item 5 below), but I also wonder if the combined index across so many components can provide a meaningful overview of the ocean state of the region. A good test is to see whether a schematic diagram could be drafted which indicates the generalised conditions of a positive/negative phase of the index. As mentioned below, the manuscript also lacks an indication of how the combined index (as well as the individual sub-indices) could be a driver of variability in the wider marine ecosystem of the region.

In addition, the choice of annual mean anomalies for some of the quantities also needs justification. The drivers of variability on shallow shelf environments can be different between winter-time and summer-time, therefore averaging across the year could be masking changes in one particular season. From the marine ecosystem impacts, consistent change in one season may be driving the variability of spawning/survival/recruitment. . . I would encourage the authors to review whether their choice of averaging periods is not masking such consistent differences in inter-annual change of the seasonal variability, and is therefore providing the most meaningful information for marine scientists researching the biogeochemical and ecosystem components of the region.

3) The lack of an index on sub-polar gyre strength: There has been no consideration of sub-polar gyre strength in any of the indices considered. Did the authors consider its inclusion? Is this basin-scale driver unimportant of the Newfoundland and Labrador Shelf region?

[Figure]
4) Figures: I must admit I very much dislike the stacked bar graph as a method of visualising the anomalies (sorry!). I find it very difficult to see the common variability (or not) across the different component indices, and would recommend the authors instead create a grid of the anomalies (see for example, the overview tables of the the ICES report on ocean climate and https://marine.gov.scot/sma/sites/default/files/omr_hadisst_temp_2018_yearlyanoms_landscape.png). The top or bottom row of such a grid could be the combined sub-index as is currently shown in figures. Figure 12 most definitely needs revising (I am not sure what the text in its caption alludes to) to consider such an approach. Such a grid would also make it more readily identifiable where the combine index is based on a smaller subset of the sub-indices due to the lack of data (see also my comment on an overview table below).

5) The Climate Index and what it means: Within marine ecosystem research, the use of single indices by researchers beyond the native discipline is attractive. Ideally, these indices are a single time series that integrate the state of the physical environment. Such indices do also however need to have a clear summary of what it means when they are positive/negative. This should be summarised in an expert statement which non-expert users can understand and refer to in their own research and publications. I will try to explain this point with an example.

For example, the North Atlantic Oscillation Index is a single time series which summarises the state of the atmosphere in the North Atlantic. The index has a clear definition (or two if you separate the calculation method into station-based and EOF-based), and this is associated with clear statements on what a positive/negative phase means for the prevailing weather conditions in North America/Northern Europe. There is a generally accepted data provider, who also provides a clear guide to non-expert users to aid the interpretation and use of this time series (see https://climatedataguide.ucar.edu/climate-data/hurrell-north-atlantic-oscillation-nao-index-pc-based). This means that any interested scientist can download the time series easily (and freely), and review the expert guidance to ensure they are aware of its applicability and limitations and use it appropriately (or not, but at their own risk). As the data provider, it does require a certain "letting go" of responsibility and ownership, as end users will take your data and use it however they see fit. It also reduces the requirement on experts to spend time liaising with each individual user on what the index time series means for their data/analysis/...

In my opinion, it is advantageous to release some of this expert guidance with the manuscript as it will aid the end user and will broaden the application the end product. As is, I think this additional expert guidance and interpretation is missing from the manuscript, and I would therefore recommend the authors consider including a "what the NLCI means for the state of the region's seas" section. I would also suggest that some of this is elaborated for each of the sub-indices too.

In the end, I do think it is up to the authors to consider inclusion of such a section. They will need to make a decision on whether they consider this a data product which is freely available but where end-users will need to make contact and collaborate to aid in the meaningful interpretation of the end-user's data, or whether this is a data product which comes with a sufficient level of expert guidance that allows end-users to make their own attempts with interpretation (but where they may still approach the authors for expertise if desired). There is no correct answer here (and different authors/reviewers will have their own bias), but the NAO Index products provide an example of what could be achieved (and how to do it well).

Other minor comments:

General: consider adding an overview table with sub-index, data source, time period covered, calculation method.

Line 16: Although an annual update is stated, the likely publication time within the year is not defined. Some end-users may want to know whether this update will be in Spring/Summer/Autumn/Winter to know whether they can expect it when they are

undertaking their own annual assessments (for example, for inclusion in an annual stock forecast for stock assessments which may be undertaken at a specific time of year). Such a statement could be appropriately "hedged" to avoid over-committing (or unexpected set-backs): "An annual update of the NLCI will likely be available by early summer each year. "

Line 27-28: It may be worth consider for the future 2020 update to create versions referenced to both the 1981-2010 and the 1991-2020 period to highlight to end-users the possible impacts of the change in reference period (or include an expert guidance statement to provide this information).

Line 51: Are these the normalised anomalies of the annual mean, or the annual mean anomalies? Line 130-131: Add reference to some of the recent papers documenting this fresh anomaly in the sub-polar North Atlantic (such as Holliday et al, 2020).

Line 143: The choice of BB as Bonavista is a little confusing, particularly as Baffin Bay is also part of the overall region, and generally abbreviated as BB.

Figure 7: Isopycnals on both panels could provide a good addition.

---

## Referee Comment (RC2) · Anonymous Referee #2 · 28 Jan 2021

The objective of this paper is to combine 10 climate subindices into a new index for describing the environmental conditions on the Newfoundland and Labrador shelf. Generally, the manuscript is well written and easy to follow. The authors devote a lot of space and figures to describing the 10 subindices. However, the significance of introducing such a new index is ambiguous. The literature review is not comprehensive and too simple. There are some critical points to be clarified. Additional assessments are also required to verify the reliability and superiority of the new index. Therefore, major revision is warranted before publication.

Specific comments:

1. The sections of Abstract and Introduction are too simple. The research gap and motivation of this study are missing. Are existing indices not good? It is desired to clarify the deficiency of previous indices, even though the authors claim that the proposed index has been used in annual reports on the physical oceanographic and meteorological conditions.

2. Line 3: The contribution of the 10 subindices is equal. Is it reasonable? Please explain.

3. The proposed climate index is introduced in a previous study by the authors and their colleagues. I find that the previous study is similar to this paper. Please explain the difference between them.

4. The authors give too many details on the 10 subindices in Section 2, but there is no statement on the methodological contribution. I am also confused about the calculation of the proposed index. How is the index calculated based on the 10 subindices?

5. Lines 181-182: "The sign of some subindices have been reversed". Is this a common procedure?

6. Figure 13 and Line 215: Figure 13 indicates that quite a few subindices are correlated significantly, so I disagree with the authors' statement that the subindices are "relatively independent".

7. Figure 14: The authors compare the proposed climate index and the CEI, and find a good correlation between them. So, what is the superiority of the proposed index? Is it better than previous indices?

8. Line 221: Why is the new index useful for ecosystem studies, fish stock assessment, and forecast models? According to the Introduction section, the new climate index is designed to better inform fisheries scientists and managers. However, there is little related evaluation in the paper. Please provide more related background or evidence.

---

## Referee Comment (RC3) · Anonymous Referee #3 · 28 Jan 2021

This study introduces a new index to represent environmental conditions of the Newfoundland and Labrador Shelf. The new index is based on 10 subindices. Having an useful index that can well depict variations of the environmental conditions is always useful and needed. The paper is well written, but I do have some questions.

(1) The winter NAO index is generally believed to have significant impacts on the hydrology in the subpolar region, but the correlations shown in Figure 13 suggest that the NAO does not strongly contribute to the variations of the other 9 subindices. I would tend to suggest the authors provide some explanation for this.

(2) Salinity from S27 appears not to be a good index to use for representing the environmental conditions of the investigated shelf waters (Figure 13). Considering the location of the station which is close to coast (by visual judgement), other factors may

have contributed to its variability rather than environmental condition changes in the shelf waters as a whole. If this subindice is removed in the calculation for the new climate index, can the performance of the new index be improved?

(3) The S27 temperature and S27 CIL have a strong correlation, which is expected. They are not really independent with each other. I am not sure how this dependency can affect this new climate index. Can the authors address this issue?

(4) Since this new climate index is intended to replace the CEI by Petrie et al. (2007), it would be helpful if the authors can provide some details about this CEI in the manuscript, which can make the paper complete.

Some detailed comments (1) Line 215 "more stable that" should be "more stable than" The paper is well written, but I would suggest the authors should pay attention to possible typos.

---

## Author Comment (AC1) · 16 Mar 2021

Dear Dr Berx,

Thank you for your exhaustive comments. Please find below a point-by-point reply to your comments. In order to better answer, the relevant part of your comments associated to our answer has been re- copied here in italic.

**1) On the calculation of some of the sub-indices**

*- Section 2.1 NAO: The choice of the EOF-based NAO means that technically the time series will be slightly different each year (due to the nature of the analysis). Similar to the information highlighted in the Hurrell product, the users of the NLCI should be*

*made aware that this means they do need to download the entire time series annually (rather than add a single value to the end of their time series). The authors could avoid this requirement by choosing the alternate NAO data product. If the current method is maintained, the caveat does need to be made explicit to ensure awareness with end-users.*

Very good point, thank you. We kept the EOF definition for consistency with other environmental studies in our group (AZMP, NAFO, etc.), but now clearly mention this caveat L.73

*- Section 2.2 Air Temperature: The inclusion of the more remote sites of Nuuk and Iqaluit needs some further explanation. How does weather at these remote sites impact ocean state on the Labrador and Newfoundland Shelf? Looking at Figure 3 (but please note my comment on the figures in point 4), it looks like the sign of anomalies at these sites is at times opposite to those more local to the Labrador and Newfoundland Shelf. The authors may want to consider giving the local and remote weather conditions separate weight in their combined index. The analysis of how the different component indices correlate may also show some stronger/clearer signals if this is done.*

The 5 sites used for air temperature have been kept because they represent both local and remote effect on the NL shelf. Also, since the stack bar plot of these 5 sites are provided, the reader can appreciate the relative consistency of all the sites in some years (generally during the coldest and warmest years), or the opposite in other years. Note that there is already more weight, however, on the NL portion with 2 sites in Newfoundland and one site in Labrador (and 4 sites out of 5 in Canada). Details on this choice have now been included L. 88 and 95.

*- Section 2.3 Sea Ice: The language around the calculation of combining the 6 times series (2 variables over 3 regions) is somewhat ambiguous. My understanding is that normalised anomalies were calculated and then averaged (arithmetic mean), correct?*

Yes, correct. The text has been re-worded L. 110.

*- Section 2.4 Iceberg Count: The region where the measurement is made should be included in the map in Figure 1.*

Done.

*- Section 2.5 Station 27: The climatological conditions at Station 27 suggest the region stratifies to some extent throughout the year (see suggestion on isopycnals for Figure 7). The vertical average temperature and salinity may therefore be masking important variability in the near-surface and near- bed layer which may be driven by different processes. The authors also describe a three-layer system at the sampling location, which to me suggests that a vertical average is possibly not the most representative of on-shelf conditions. Could S27 surface and near-bed salinity and temperature be treated separately?*

We gave a lot of thought to this. In an earlier version of the NLCI, the anomalies of the near-surface, mid-water column and bottom were calculated before being averaged. The two methods did not lead to significant changes to the NLCI. We thus decided to go for the simple approach of averaging the whole water column. This is also the historical time series reported annually in the ICES Report on Ocean Climate (https://ocean.ices.dk/core/iroc).

*- Section 2.5 Station 27: The complete lack of significant correlation of the S27 salinity is one indication to me that the choices here should be reconsidered. [...] Is one of these more/less relevant for the ecosystem of the region (for example, are there known links between fish stocks and recruitment success and one/several of these climate variables?)?*

Yes, salinity has been related, for example, to capelin dynamics, although no clear causal effect has been established (a hypothesis is through stratification and the timing of the spring bloom, but this is outside the scope of this study). But more importantly, salinity at Station 27 is a good indication of the freshwater fluxes on the Labrador shelf, one of the key pathways for Arctic freshwater exports. Therefore salinity has been kept for these reasons, but users can construct their own index and remove it since all subindices are provided. This is now explained in the Discussion L.260.

*- Section 2.5 Station 27: Could the strength of stratification (for example expressed as potential energy anomaly) or the size of salinity/temperature range in the year be more important (on lines 115-117 the authors highlight the importance of the salinity cycle, but this is not adequately reflected in the sub- index or eventual climate index). Generally, salinity is a good indicator of circulation change, and therefore I would have suspected it to play a more important role, particularly due to the sub-polar gyre's influence on the region (see also item 3 below).*

This is a very good point, but such analysis seems outside the scope of this study and would not be in line with the previous versions of the NL climate indices.

*- Section 2.8 Bottom temperature: What is the reason for choosing the 1000 m isobath to delimit the extent of the shelf? Most publications consider the boundary to be the 200*
*or 500 m isobaths. Is it because this is what the fisheries assessments use? Does this definition mean that a significant portion of water masses from deeper in the Labrador Sea is included? How does this inclusion of deeper water (which is likely not influence by the same processes as the shallow shelf region) significantly impact on the bottom temperature mean and its seasonal/inter-annual variability?*

Selecting 200m would be too shallow for the different channels on the shelf. While 500m might be a suitable choice, we would lose some areas in the deep Laurentian Channel in the south of our region. The isobath 1000m has then been chosen because some NL fisheries extent to the shelf break near these depths and because some oceanographic data that we used are limited to 1200m profile depths.

**2) On the oceanographic understanding behind the combined index**

*This index will be very valuable to other marine scientists studying the ecosystem dynamics and productivity of the region. There is little interpretation of this throughout the manuscript (see item 5 below), but I also wonder if the combined index across so many components can provide a meaningful overview of the ocean state of the region. A good test is to see whether a schematic diagram could be drafted which indicates the generalised conditions of a positive/negative phase of the index. As mentioned below, the manuscript also lacks an indication of how the combined index (as well as the individual sub-indices) could be a driver of variability in the wider marine ecosystem of the region.*

In a hope to strengthen our explanation on the interpretation and the role of the NLCI for the ecosystem, and the interactions between the different subindices, we have expanded the Discussion. While some scientists already find meaningfulness with the NLCI (see review in the Introduction),

we also provide the 10 subindices, so users can design their own index. While a schematic diagram would be an interesting addition, the amount of work required seems unrealistic as part of this review (especially that we already have 14 figures). Such a description could well be a dedicated study by itself.

*In addition, the choice of annual mean anomalies for some of the quantities also needs justification. The drivers of variability on shallow shelf environments can be different between winter-time and summer-time, therefore averaging across the year could be masking changes in one particular season. From the marine ecosystem impacts, consistent change in one season may be driving the variability of spawning/survival/recruitment... I would encourage the authors to review whether their choice of averaging periods is not masking such consistent differences in inter-annual change of the seasonal variability, and is therefore providing the most meaningful information for marine scientists researching the biogeochemical and ecosystem components of the region.*

We agree on this, and we now make this distinction in the Conclusion L.278. While it is difficult to have one single index that is representative of all seasons, our approach is to provide 10 subindices where some represent more the winter season (NAO, sea ice), some more the ice-free season (SST, CIL) and some the entire year (air temperature, S27 T-S). While the combination of all subindices in the NLCI is already in high demand among our colleagues, this new version with available 10 subindices is a significant addition for future ecosystem studies.

*3) The lack of an index on sub-polar gyre strength: There has been no consideration of sub-polar gyre strength in any of the indices considered. Did the authors consider*

*its inclusion? Is this basin-scale driver unimportant of the Newfoundland and Labrador Shelf region?*

This is a good point, but available data for a "Labrador Current subindex" would only starts in 1993 which is quite limiting to our time scale that aims to capture decadal variations. In addition, such an addition would differ with previous versions of the NL climate indices.

*4) Figures: I must admit I very much dislike the stacked bar graph as a method of visualising the anomalies (sorry!). I find it very difficult to see the common variability (or not) across the different component indices, and would recommend the authors instead create a grid of the anomalies [...] Such a grid would also make it more readily identifiable where the combine index is based on a smaller subset of the sub-indices due to the lack of data (see also my comment on an overview table below).*

We have modified Figure 12 to add such scorecards.

*5) The Climate Index and what it means: Within marine ecosystem research, the use of single indices by researchers beyond the native discipline is attractive. Ideally, these indices are a single time series that integrate the state of the physical environment. Such indices do also however need to have a clear summary of what it means when they are positive/negative. This should be summarised in an expert statement which non-expert users can understand and refer to in their own research and publications. I will try to explain this point with an example. [...] In my opinion, it is advantageous to release some of this expert guidance with the manuscript as it will aid the end user and will broaden the application the end product. As is, I think this additional expert guidance and interpretation is missing from the manuscript, and I would therefore recommend the authors consider including a "what the NLCI means for the state of the*

*region's seas" section. I would also suggest that some of this is elaborated for each of the sub-indices too. In the end, I do think it is up to the authors to consider inclusion of such a section. They will need to make a decision on whether they consider this a data product which is freely available but where end-users will need to make contact and collaborate to aid in the meaningful interpretation of the end-user's data, or whether this is a data product which comes with a sufficient level of expert guidance that allows end-users to make their own attempts with interpretation (but where they may still approach the authors for expertise if desired). There is no correct answer here (and different authors/reviewers will have their own bias), but the NAO Index products provide an example of what could be achieved (and how to do it well).*

This comment looks like no.2 above. We have re-written the Discussion in hopes of providing a basic expert guidance on the NLCI. A complete description on the functioning of the NL shelf ecosystem in relation with the NW Atlantic conditions is however still a work in progress, and outside the scope of this study.

**Other minor comments:**

*General: consider adding an overview table with sub-index, data source, time period covered, calculation method.*

We now provide a table with the historical versions of the climate indices in Atlantic Canada.

*Line 16: Although an annual update is stated, the likely publication time within the year is not defined. Some end-users may want to know whether this update will be in Spring/Summer/Autumn/Winter to know whether they can expect it when they are undertaking their own annual assessments (for example, for inclusion in an annual*

*stock forecast for stock assessments which may be undertaken at a specific time of year). Such a statement could be appropriately "hedged" to avoid over-committing (or unexpected set-backs): "An annual update of the NLCI will likely be available by early summer each year. "*

Such a sentence has been added in the data availability statement.

*Line 27-28: It may be worth consider for the future 2020 update to create versions referenced to both the 1981-2010 and the 1991-2020 period to highlight to end-users the possible impacts of the change in reference period (or include an expert guidance statement to provide this information).*

This new version of the manuscript is now based in the 1991-2020 climatolgoy. A suite of relevant figures using the 1981-2010 climatology is also provided in the Appendix.

*Lines 130-131: Add reference to some of the recent papers documenting this fresh anomaly in the sub-polar North Atlantic (such as Holliday et al, 2020).*

Done L.182

*Line 143: The choice of BB as Bonavista is a little confusing, particularly as Baffin Bay is also part of the overall region, and generally abbreviated as BB.*

BB stands for Bonavista Bay (now stated in the text) And has been kept for historical reason (it has been sampled and named with this acronym since the 1950's)

*Figure 7: Isopycnals on both panels could provide a good addition.*

Done

---

## Author Comment (AC2) · 16 Mar 2021

Dear Reviewer,

Thank you for your comments. Please find below a point-by-point reply to your comments. In order to better answer, the relevant part of your comments associated to our answer has been re-copied here in italic.

*The objective of this paper is to combine 10 climate sub-indices into a new index for describing the environmental conditions on the Newfoundland and Labrador shelf. Generally, the manuscript is well written and easy to follow. The authors devote a lot of space and figures to describing the 10 subindices. However, the significance of intro-*

*ducing such a new index is ambiguous. The literature review is not comprehensive and too simple. There are some critical points to be clarified. Additional assessments are also required to verify the reliability and superiority of the new index. Therefore, major revision is warranted before publication.*

**Specific comments:**

*1. The sections of Abstract and Introduction are too simple. The research gap and motivation of this study are missing. Are existing indices not good? It is desired to clarify the deficiency of previous indices, even though the authors claim that the proposed index has been used in annual reports on the physical oceanographic and meteorological conditions.*

> We acknowledge this aspect. The Introduction has been completely rewritten (and the abstract amended). This new index is produced because the previous one had been abandoned when one of our colleagues retired. The context in which this new index has been developed has now been clarified and a new Table 1 now details the different historical versions of these indices.

*2. Line 3: The contribution of the 10 subindices is equal. Is it reasonable? Please explain.*

> We generate the index in the simplest possible way: the arithmetic average of the 10 subindices. We have no justification to make it otherwise, but by providing the 10 subindices, users can generate their own custom index if needed.

*3. The proposed climate index is introduced in a previous study by the authors and their colleagues. I find that the previous study is similar to this paper. Please explain the difference between them.*

The previous study mentioned above is not peer-reviewed in an international journal and few details are given about this climate index. In addition, the climate index presented here has been changed compared to the one presented previously. Differences now explained in the Introduction (see L.49).

*4. The authors give too many details on the 10 subindices in Section 2, but there is no statement on the methodological contribution. I am also confused about the calculation of the proposed index. How is the index calculated based on the 10 subindices?*

Simple arithmetic average of the 10 normalized anomalies (see L. 230)

*5. Lines 181-182: "The sign of some subindices have been reversed". Is this a common procedure?*

Yes, this is common practice because anomalies would otherwise cancel each other out when averaged. For example, a negative sea ice anomaly implies warm conditions, and thus need to be reversed to match, for example, warm positive/warm air temperature anomalies. Note that the subindices are also provided in their natural sign in the dataset. We are sorry if this was not clear, we have modified the text near L.228.

*6. Figure 13 and Line 215: Figure 13 indicates that quite a few subindices are correlated significantly, so I disagree with the authors' statement that the subindices are "relatively independent".*

We acknowledge that these subindices are correlated (see discussion starting in L.235). What we mean with this statement is that since no correlation

greater than 0.9 is found, each subindex captures a different part of the variance of the NLCI. This is contrary to the previous versions of the climate index where repetition was found in the 28 subindices (several air temperatures, several SSTs, several bottom temperatures, etc.). We now make this clear L.270.

*7. Figure 14: The authors compare the proposed climate index and the CEI, and find a good correlation between them. So, what is the superiority of the proposed index? Is it better than previous indices?*

The goal of this study was not to demonstrate a superiority from a previous version of the index, but rather to ensure continuity from a previously abandoned index after the retirement of a colleague. In addition, the goal here is to make this new index stable, fully transparent and open access. We see a good correlation with the previous version(s) as a good thing (for continuity reasons), given that this new index is simpler (only 10 sub-indices rather than 28). We hope that this study will also give some stability to this "NL climate index" for which the definition has been constantly changing over the past 15 years. Finally, this index is also based on more modern data (e.g. better ice product, better SST product, etc.).

*8. Line 221: Why is the new index useful for ecosystem studies, fish stock assessment, and forecast models? According to the Introduction section, the new climate index is designed to better inform fisheries scientists and managers. However, there is little related evaluation in the paper. Please provide more related background or evidence.*

We have completely re-written and augmented the Introduction to address this comment.

---

## Author Comment (AC3) · 16 Mar 2021

Dear Reviewer, Thank you for your comments. Please find below a point-by-point reply to your comments. In order to better answer, the relevant part of your comments associated to our answer has been re-copied here in italic
* * *
*This study introduces a new index to represent environmental conditions of the Newfoundland and Labrador Shelf. The new index is based on 10 subindices. Having a useful index that can well depict variations of the environmental conditions is always useful and needed. The paper is well written, but I do have some questions.*

*(1) The winter NAO index is generally believed to have significant impacts on the hydrology in the subpolar region, but the correlations shown in Figure 13 suggest that the NAO does not strongly contribute to the variations of the other 9 sub-indices. I would tend to suggest the authors provide some explanation for this.*

We see this lack of strong correlation as rather a good thing. It means that the NLCI captures a more complex dynamics that is not entirely captured by the NAO alone. While the winter NAO captures some decadal dynamics (warm 60's, cold 90's; warm 2010's, etc.), the year-to-year correlation between all subindices is likely weaker because of lag/inertia effects. We now discuss this near L.250.

*(2) Salinity from S27 appears not to be a good index to use for representing the environmental conditions of the investigated shelf waters (Figure 13). Considering the location of the station which is close to coast (by visual judgement), other factors may have contributed to its variability rather than environmental condition changes in the shelf waters as a whole. If this subindice is removed in the calculation for the new climate index, can the performance of the new index be improved?*

Station 27 is located within the Avalon channel, which is one of the main pathways of Arctic-origin waters flowing along the Labrador coast. Previous studies have shown that the geographical location of the Station is very relevant for the NL shelf as a whole [e.g. Petrie et al.: Temperature and salinity variability on the eastern Newfoundland shelf: The annual harmonic, Atmosphere-Ocean, 29, 14–36, https://doi.org/10.1080/07055900.1992.9649433, 1991]. While the correlation between salinity and the NLCI is not significant, salinity has been related, for example, to capelin dynamics, although no clear causal effect has been established. We thus believe that is it important to keep salinity for
these reasons, but users can remove it from the NLCI since all subindices are provided. This is now explained in the Discussion L.260.

*(3) The S27 temperature and S27 CIL have a strong correlation, which is expected. They are not really independent with each other. I am not sure how this dependency can affect this new climate index. Can the authors address this issue?*

Yes, a large part of the Stn27 temperature is driven by the CIL temperature, but we have chosen to only consider the "core temperature" of the CIL (minimum temperature of the profile) for which the correlation is 0.72. This signifies part of the variance in one is not captured by the other. The vertically averaged temperature at Station 27 also includes surface and bottom temperature that are not always in phase with the CIL core temperature.

*(4) Since this new climate index is intended to replace the CEI by Petrie et al. (2007), it would be helpful if the authors can provide some details about this CEI in the manuscript, which can make the paper complete.*

An historical review of previous studies presenting such climate index is now included in the Introduction. A table presenting an overview of these studies and its components is also provided (Table 1)

---

## Author Comment (AC4) · 16 Mar 2021

Dear Editor,

We are pleased to provide a new version of our manuscript entitled "A climate index for the Newfoundland and Labrador shelf" which we hope to publish in ESSD. We have carefully addressed the comments from the 3 reviewers. Among important changes, the NL climate index (NLCI) was extended to include the year 2020. Consequently, a new climatological period (1991-2020) was used. The relevant figures to the NLCI and its subindices have also been provided using the 1981-2010 climatology in a new appendix. Another important change is a complete re-writing of the Introduction and the Discussion following comments made from the reviewers.

Finally, since the submission of the data set and the apparition of our manuscript in the Discussion Forum of ESSD, the climate index had a great success among our colleagues. As of March 15th , the manuscript and the data set have been seen 266 and 108 times, respectively. Both were submitted mid-November 2020. We are confident that this early enthusiasm for our study and the climate index indicates a high citation potential for the journal.

Best Regards,

Frederic Cyr and Peter S. Galbraith